# Modelling spatiotemporal patterns of visceral leishmaniasis incidence in two endemic states in India using environment, bioclimatic and demographic data, 2013–2022

**Swaminathan Subramanian**[1]\*, **Rajendran Uma Maheswari**[1],
**Gopalakrishnan Prabavathy**[1], **Mashroor Ahmad Khan**[1], **Balan Brindha**[1],
**Adinarayanan Srividya**[1], **Ashwani Kumar**[1], **Manju Rahi**[1,2], **Emily S. Nightingale**[3], **Graham
F. Medley**[3], **Mary M. Cameron**[4], **Nupur Roy**[5], **Purushothaman Jambulingam**[1]

**1** ICMR-Vector Control Research Centre, Indira Nagar, Puducherry, India, **2** Division of Epidemiology and Communicable Diseases, Indian Council of Medical Research, New Delhi, India, **3** Centre for Mathematical Modelling of Infectious Disease and Department of Global Health and Development, London School of Hygiene and Tropical Medicine, London, United Kingdom, **4** Department of Disease Control, London School of Hygiene and Tropical Medicine, London, United Kingdom, **5** National Centre for Vector-Borne Diseases Control, Ministry of Health and Family Welfare, Government of India, New Delhi

☯ These authors contributed equally to this work.
\* ssubra@yahoo.com

**Data Availability Statement:** The data from the Kala-Azar Management Information System

## Abstract

### Background

As of 2021, the National Kala-azar Elimination Programme (NKAEP) in India has achieved visceral leishmaniasis (VL) elimination (<1 case / 10,000 population/year per block) in 625 of the 633 endemic blocks (subdistricts) in four states. The programme needs to sustain this achievement and target interventions in the remaining blocks to achieve the WHO 2030 target of VL elimination as a public health problem. An effective tool to analyse programme data and predict/ forecast the spatial and temporal trends of VL incidence, elimination threshold, and risk of resurgence will be of use to the programme management at this juncture.

### Methodology/principal findings

We employed spatiotemporal models incorporating environment, climatic and demographic factors as covariates to describe monthly VL cases for 8-years (2013–2020) in 491 and 27 endemic and non-endemic blocks of Bihar and Jharkhand states. We fitted 37 models of spatial, temporal, and spatiotemporal interaction random effects with covariates to monthly VL cases for 6-years (2013–2018, training data) using Bayesian inference via Integrated Nested Laplace Approximation (INLA) approach. The best-fitting model was selected based on deviance information criterion (DIC) and Watanabe-Akaike Information Criterion (WAIC) and was validated with monthly cases for 2019–2020 (test data). The model could describe observed spatial and temporal patterns of VL incidence in the two states having widely differing incidence trajectories, with >93% and 99% coverage probability (proportion of

(KAMIS) underlying the results in this manuscript cannot be shared publicly because of patient confidentiality and privacy concerns. KA-MIS data are property of the National Vector-Borne Disease Control Programme (NVBDCP, Govt of India), and are managed by CARE India. The data are available from NVBDCP (nvbdcp-mohfw@nic.in) for researchers who meet the criteria for access to confidential data.

**Funding:** This study was supported by the Bill and Melinda Gates Foundation (https://www.gatesfoundation.org/) through the SPEAK India consortium [OPP1183986] (SS, RU, GP, MAK, BB, ASV, AK, ESN, MMC, GFM, PJ). The views, opinions, assumptions or any other information set out in this article are solely those of the authors and should not be attributed to the funders or any person connected with the funders. The funders had no role in study design, data collection and analysis, decision to publish, or preparation of the manuscript.

**Competing interests:** The authors have declared that no competing interests exist.

observations falling inside 95% Bayesian credible interval for the predicted number of VL cases per month) during the training and testing periods. PIT (probability integral transform) histograms confirmed consistency between prediction and observation for the test period. Forecasting for 2021–2023 showed that the annual VL incidence is likely to exceed elimination threshold in 16–18 blocks in 4 districts of Jharkhand and 33–38 blocks in 10 districts of Bihar. The risk of VL in non-endemic neighbouring blocks of both Bihar and Jharkhand are less than 0.5 during the training and test periods, and for 2021–2023, the probability that the risk greater than 1 is negligible (P<0.1). Fitted model showed that VL occurrence was positively associated with mean temperature, minimum temperature, enhanced vegetation index, precipitation, and isothermality, and negatively with maximum temperature, land surface temperature, soil moisture and population density.

## Conclusions/significance

The spatiotemporal model incorporating environmental, bioclimatic, and demographic factors demonstrated that the KAMIS database of the national programmme can be used for block level predictions of long-term spatial and temporal trends in VL incidence and risk of outbreak / resurgence in endemic and non-endemic settings. The database integrated with the modelling framework and a dashboard facility can facilitate such analysis and predictions. This could aid the programme to monitor progress of VL elimination at least one-year ahead, assess risk of resurgence or outbreak in post-elimination settings, and implement timely and targeted interventions or preventive measures so that the NKAEP meet the target of achieving elimination by 2030.

### Author summary

In India, VL has been endemic in four states (Bihar, Jharkhand, Uttar Pradesh, and West Bengal), having over 165 million population. The national programme achieved elimination (<1 case / 10,000 population/year per 'block') in 625 of the 633 endemic blocks in 2021. While sustaining elimination level, the programme needs to target other blocks yet to reach elimination to achieve the WHO 2030 target. We fitted a variety of spatiotemporal models to 72-monthly reported VL cases (2013–2018, training period) from 491 endemic and 27 non-endemic blocks in Bihar and Jharkhand. The best fitting model was validated with 24-month reported cases (2019–2020, test period). Model predictions agree with >93 and 99% of the monthly-observations for the periods. Forecasting for 2021–2023 showed that incidence is likely to exceed elimination threshold in 16–18 and 33–38 historically high endemic blocks of Jharkhand and Bihar. Fitted model showed that VL incidence is positively associated with mean temperature, minimum temperature, enhanced vegetation index, precipitation, and isothermality, and negatively with maximum temperature, land surface temperature, soil moisture and population density. Forecasting VL incidence at block level can aid to monitor elimination progress, target the blocks yet to reach elimination and long-term monitoring of risk of resurgence during post-elimination.

## 1. Introduction

Leishmaniases are neglected tropical diseases caused by Leishmania protozoa transmitted to humans by the bites of infected female phlebotomine sand flies [1]. In India, visceral leishmaniasis (VL) is caused by *Leishmania donovani* and transmitted by the infective bites of *Phlebotomus argentipes*. As of 2021, over 165 million people [2] from 633 blocks (subdistricts) in 54 districts of the four endemic states, Bihar (458/534 blocks in 33 districts), Jharkhand (33/34 blocks in 4 districts), Uttar Pradesh (22/133 blocks in 9 districts) [3] and West Bengal (120/204 blocks in 11 districts) remain at risk of VL infection. The National Health Policy-2002 set the goal of kala-azar elimination as a public health problem (less than 1 case per 10,000 population per year at block level) in India by the year 2010, which was revised to 2015 and again to 2030. The National Kala-Azar Elimination Programme (NKEP) has aimed to achieve the goal of elimination of VL by detection and treatment of cases and reducing vector density by indoor residual spraying with synthetic pyrethroids. The programme has made significant progress and achieved elimination in 625 of the 633 endemic blocks in the four states. Only 8 blocks in 6 districts were above the elimination threshold in 2021 [2]. The disease continued to decline from 29,000 cases in 2010 to less than 2000 cases in 2021 [4,5]. Under NKEP it is mandatory for all the states to report kala-azar cases every month, including 'zero' cases, to a repository centrally administered by National Vector-Borne Diseases Control Programme (NVBDCP, now renamed as 'National Centre for Vector Borne Diseases Control, NCVBDC), Ministry of Health and Family Welfare (MOHFW), Government of India, called Kala-Azar Management Information System (KAMIS). The KAMIS database is being used to monitor the spatial and temporal trends in VL incidence. The data can further be potentially used to predict / forecast VL outbreak or resurgence especially during post elimination. This is important to rule out the possibility that the observed down-trend being accelerated by the "natural" fluctuation of the disease (disease incidence in India is cyclic) [6,7] rather than entirely due to the effect of interventions [8], as well as to prevent potential outbreaks when herd immunity is in weakening phase [9].

There has been growing interest on the application of statistical–spatial, temporal and spatiotemporal models to quantify spatial, temporal and spatiotemporal patterns of vector-borne diseases, e.g. lymphatic filariasis [10,11], malaria [12–15], dengue [16] and health outcomes [17]. While spatial models account for correlation and draw strength across neighbouring areas to produce more stable estimates of disease risk (areas close together tend to have more similar risks than areas far apart) [18], temporal models account for correlation over time lag and indicate how risk evolves over time. These models are valuable to assess the effect of prevention or intervention measures. However, investigating only spatial or temporal pattern of disease may not able to demonstrate how a disease is changing over time and space. Spatiotemporal models overcome these limitations by enabling identification of how disease risk varies over both space and time.

A few studies in the countries endemic for cutaneous leishmaniasis (CL) (Costa Rica, Colombia, Brazil, and Sri Lanka) have investigated association of spatial and temporal distribution of CL incidence with climate and environment variables [19–25]. These studies have applied either spatial models or geostatistical models to determine the predictors of CL incidence. To our knowledge, only two studies in Brazil [26] and Ethiopia [27], have applied spatiotemporal models to identify the underlying risk factors of human VL incidence. The study in Brazil used GAM (generalized additive models) and showed that quality of life in an urban area (a composite index related to income, education, housing, and environmental sanitation) was inversely related to VL incidence. Godana et al. [27] in Ethiopia followed a stochastic partial differential equation approach using Integrated Nested Laplace Approximation (INLA)

and confirmed the association of VL incidence with meteorological, demographic, sociodemographic, and geographic covariates at health facility level.

Statistical scanning techniques have been employed in Bangladesh to describe the spatial-temporal heterogeneity or clustering of VL cases at regional scale. However, the study neither considered any risk factors to identify the clusters nor to forecast incidence [28]. In India, analyses of the association of VL incidence with climatic factors, and vector density have been done but are mostly limited to small geographical areas [9,29–31]. The studies focused towards identifying drivers of hotspots at the village or household level [9,30,31]. Deb et al. [7], applied a negative binomial regression model to the state level annual VL incidence data from Bihar and showed significant negative associations of VL incidence with maximum temperature, and average temperature. Bhunia et al. [29], analyzed district level VL incidence data and observed VL incidence in the Gangetic plain of Bihar is positively associated with environmental (presence of water bodies, woodland and urban, built-up areas, soil type) and climatic (air temperature, relative humidity and annual rainfall) factors.

Nightingale et al. [32], applied a spatiotemporal model to the block level monthly VL cases reported in Bihar and Jharkhand from 2013–2018, and showed that VL incidence at block level can be predicted three or four months ahead with similar accuracy and precision as one-month ahead. However, the analysis did not consider other factors that could have potentially impacted the spatial and temporal patterns of VL incidence. The population dynamics of sand flies, the vectors of VL, depend on environmental, demographic, and human behavioral factors, and hence the diseases caused by leishmania parasites and transmitted by sandflies are dynamic [33]. With the elimination of visceral leishmaniasis targeted for 2030, it is necessary to understand the spatial and temporal dynamics of the disease incidence and its relationship with potential risk factors so that preventive / effective control strategies can be targeted appropriately.

In this paper, we extend the work of Nightingale et al. [32] by incorporating potential risk factors (environment, bioclimatic and demographic covariates) of VL incidence into spatial, temporal and spatiotemporal models, using the Integrated Nested Laplace Approximation (INLA), which is computationally less challenging than MCMC (Markov Chain Monte Carlo) methods for Bayesian inference. We validate the best fitting model with observations not included in model building and predict the incidence trends beyond the period of observations.

## 2. Methods

### 2.1. Ethics statement

The VL case data were collected as part of routine programme activities conducted by the NCVBDC and therefore no ethical clearance was required for secondary data analysis. The ICMR-VCRC (Indian Council of Medical Research–Vector Control Research Centre) has obtained approval to use the secondary data from the NCVBDC and the proposal was cleared by the Health Ministry Clearance Committee, Govt of India. Ethical clearance was also obtained from the Observational/Interventions Research Ethics Committee at London School of Hygiene and Tropical Medicine (LSHTM) (ref: 14674). As all data were analyzed anonymously, individual consent was not required.

### 2.2. Study area

Fig 1 shows the map of study areas (Bihar and Jharkhand states) and their geographic location in India. Bihar is a state located in eastern India, bordered by Nepal, West Bengal, Uttar Pradesh and Jharkhand. According to 2011 Census, it is the third-largest state by population and

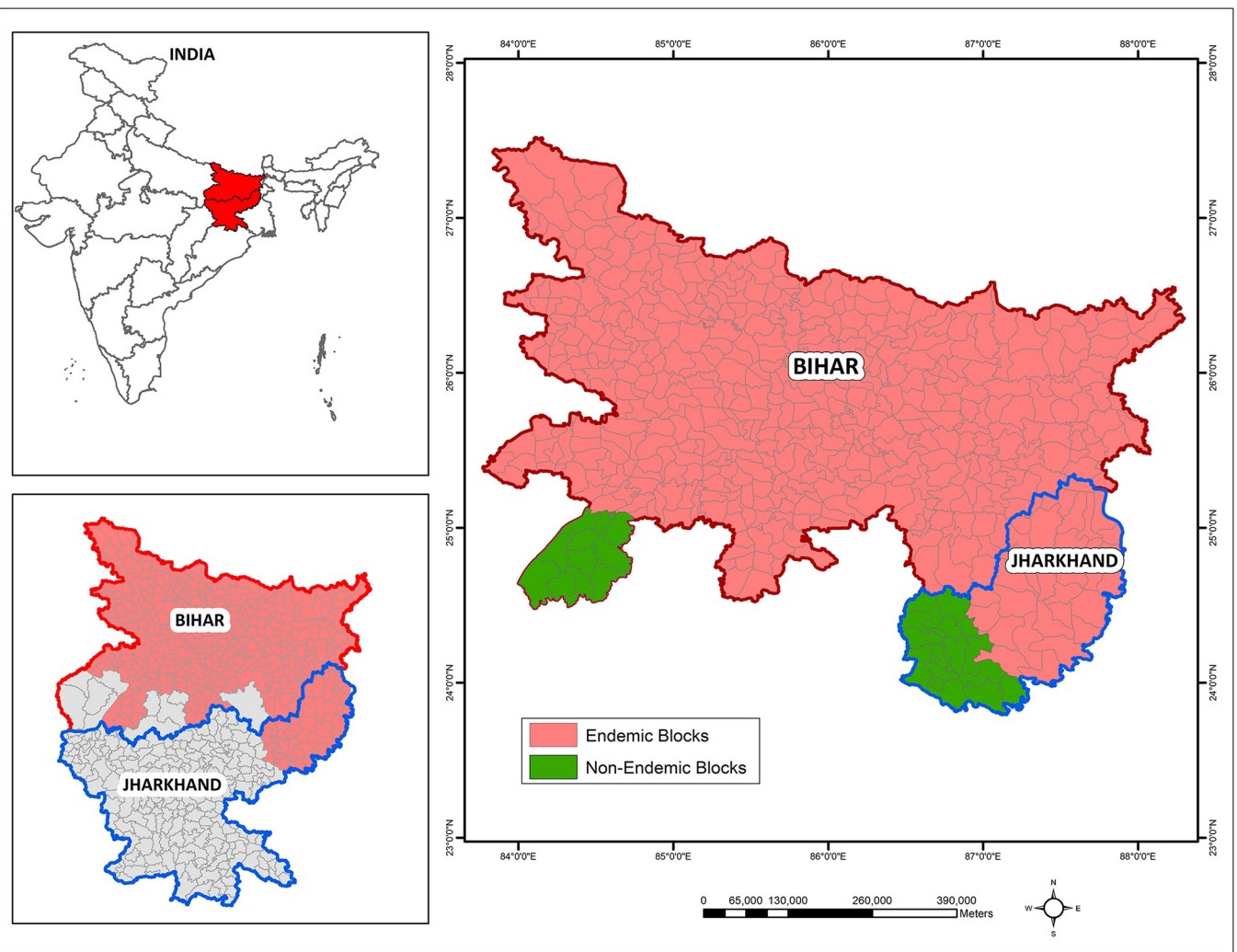

**Fig 1. Map of study area and its geographical location in India.** Endemic blocks (peach shade) in the states of Bihar and Jharkhand along with non-endemic blocks (green shade) bordering endemic blocks in the two states. The base layer map for India state boundary were downloaded from https://onlinemaps.surveyofindia.gov.in. Block level shapefile for Bihar and Jharkhand were developed in ArcGIS software (https://www.arcgis.com) by digitization tool using base layer from the India village directory, Census of India 2011, download from https://lgdirectory.gov.in.

the 12[th] largest by land which lies between 21˚58'02"N to 25˚08'32"N latitude and 83˚19'05"E to 87˚55'03"E, longitude with an average elevation above sea level of 173 feet (53 m). The state is divided into 38 districts, 534 blocks, 8471 panchayats, and 45,103 villages for administrative purposes. Bihar has a total land area of 94,163 km$^2$, with 91,838.28 km$^2$ (97.5%) being rural and 2,324.72 km$^2$ (2.5%) being urban.

Jharkhand is another state located in eastern India, adjacent to Bihar. It was carved out of the southern part of Bihar on 15 November, 2000. Jharkhand shares its border with the states of Bihar to the north, Uttar Pradesh and Chhattisgarh to the west, Odisha to the south, and West Bengal to the east. The state has a land area of approximately 79,714 km$^2$ and is located between latitude 21˚57' N to 25˚14' N and longitude 83˚ 19' 05" E to 87˚ 55' 03" E. Jharkhand has 24 districts and 260 blocks, 32,615 revenue villages, and a population of 33 million people, making it the 13[th] most populous state in India.

A block is the second level administrative unit after a 'district', and there are several blocks in each district. Fig 1 depicts the map of the blocks in the states of Bihar and Jharkhand under investigation. The blocks were chosen as the unit of analysis as the national programme aims to achieve elimination at block level, defined as 1 VL case per 10,000 population per year at block level.

According to the 2011 Census data, 88.7% of population in Bihar live in rural blocks, while 11.3% of population reside in urban blocks of Bihar. The population density in the state is 1102.4 people per $km^2$, with 1005.4 in rural (range: 829 to 40801) and 5057.8 in urban blocks (range: 62 to 3056). In Jharkhand around 76% of population lives in rural blocks and the balance of 24% in urban blocks. The state level population density is about 414 per $km^2$, with 323.4 in rural (range: 60 to 1955) and 3527.5 per $km^2$ in urban (range: 539 to 29017) blocks.

For the present study, we have included 469 of the 534 blocks in the state of Bihar (all the 458 VL endemic blocks spread over 33 districts, and 11 non-endemic blocks bordering the endemic blocks in Aurangabad district (Fig 1). Of those included, 330 blocks (includes 11 non-endemic) are rural settlements with populations ranging from 30777 to 4,35,676 and 139 being a mix of urban and rural settlements with populations ranging from 4406 to 16,87,828. The population density ranged from 199.3 to 10512.4/$Km^2$ in different blocks. The total number of non-endemic VL blocks in Bihar was 65, with 63 being rural and 2 being urban settlements.

In Jharkhand state, we have included a total of 49 of the 260 blocks (all the 33 VL endemic blocks in 4 districts and 16 non-endemic blocks bordering the endemic blocks were chosen from Jamtara and Deoghar districts) (Fig 1). Out of 49 blocks, 14 blocks are occupied by mixed urban and rural villages with populations ranging from 5868 to 2,03, 123, while the remaining 35 blocks (includes 16 non-endemic blocks) are rural blocks with population ranging from 42,063 to 2,60,403. The population density in the state ranged from 377.3 to 1393.9/$km^2$ in different blocks. In Jharkhand, the total number of non-endemic blocks was 227, with 207 being rural and 20 being urban settlements.

## 2.3. Data

**2.3.1. VL case data.**   This study used the block level monthly VL cases reported from the two VL endemic states, namely Bihar and Jharkhand. The data were collected by the NCVBDC facilitated by CARE India using KAMIS. KAMIS data base has details of individual case records by date of diagnosis and geo-coordinates. We downloaded individual case records for Bihar and Jharkhand for the period from 01 January 2013 to May 2021, which were then aggregated by block, and month of diagnosis. The block wise monthly data were then merged with environment, bioclimatic and demographic data set.

**2.3.2. Covariates data.**   A description of the block-level covariates considered in this study to evaluate their possible association with VL incidence is given in S1 Table.

## Environment, climate and bioclimatic data

We have extracted remotely sensed monthly covariates data available for eight variables, for each of 518 endemic and non-endemic blocks (Bihar: 458 endemic and 11 non-endemic blocks in 33 districts; Jharkhand: 33 endemic blocks in 4 districts and 16 non-endemic blocks in 2 districts) for the period from Jan 2013 to Mar 2023. The eight variables are related to bioclimatic, environment and climatic factors. The bioclimatic variables include BIO-1(mean temperature), BIO-3 (isothermality) and BIO-12 (precipitation) with spatial resolution of 1 km were extracted from the WorldClim database (ver. 2.1, released in Jan 2020, https://www.worldclim.org/) (S1 Table) using a 'Biovar' function in R. The environment variables such as

land surface temperature (LST, 6 km resolution), enhanced vegetation index (EVI, 1 km resolution) were extracted from MODIS (Moderate-resolution Imaging Spectroradiometer) data base using 'MODISTools' package in R [34] https://github.com//seantuck12/MODISTools) and soil moisture (0.25˚ x 0.25˚ resolution) extracted from Copernicus (https://cds.climate.copernicus.eu/). The two climatic variables, maximum and minimum temperature (1 km resolution) were extracted from WorldClim database. Block level estimates for each covariate were obtained by specifying the latitude and longitude of a block representing the centroid.

As the covariates data were available only up to March 2023, the data for the period April 2023 –December 2023 were derived by averaging the values for each covariate per month over the period from April 2013—Dec 2022.

## Demographic data

Block wise population data and decadal growth rates for 2011 were obtained directly from the 2011 national census [35]. The block level population data for the period from January 2013 to December 2023 were estimated according to estimated decadal growth rates for each block. The estimated population per month for each block was then used to calculate the population density per $km^2$.

## 2.4. Statistical analysis

**2.4.1. Selection of covariates.**   We calculated the Pearson's correlation coefficient (r)) for each year (temporal correlation) to identify the covariates that significantly correlated with monthly VL incidence (combined for all blocks). Similarly, 'r' was also calculated for each block to assess the spatial correlation between covariates and monthly VL incidence (combined for all years). Covariates with $P < 0.2$ at least in one of the years during 2013–2020 or in one of the blocks are considered for subsequent Bayesian spatiotemporal modelling analysis.

**2.4.2. Standardized incidence ratio (SIR).**   We calculated the SIR in block $i$ ($i = 1, \ldots,$ 518) and month $j$ (1, 2, . . .., 12) in year $t$ ($t = 2013, 2014, \ldots, 2020$) as the ratio of the number of observed cases $O_{ijt}$ to the number of expected cases $E_{ijt}$ in the $i^{th}$ block in month $j$ and year $t$:

$$SIR_{ijt} = \frac{O_{ijt}}{E_{ijt}},$$

with the expected number of cases calculated as,

$$E_{ijt} = N_{ijt} \times r^{2013},$$

Where $N_{ijt}$ is the population in the $i^{th}$ block in month j and year t, and $r^{2013}$ is the reference rate based on all blocks and months in the base year 2013 and is calculated as,

$$r^{2013} = \frac{\sum_{j=1}^{12} \sum_{i=1}^{518} O_{ij}^{2013}}{\sum_{j=1}^{12} \sum_{i=1}^{518} N_{ij}^{2013}}$$

Where $O_{ij}^{2013}$ and $N_{ij}^{2013}$ are the reported number of VL cases and projected census population in block $i$, in month $j$ year 2013 respectively.

Areas with '$SIR$' values higher than 1 will stand for an excess of risk, while values lower than 1 mean a lower risk for the population in that unit. However, these measures are extremely variable when analyzing rare diseases or low-populated areas, as is the case of high-dimensional data. In order to manage this situation, statistical models that stabilize the risks (rates) borrowing information from neighboring regions are being considered. Generalized linear

mixed models (GLMM) are typically used for the analysis of count data within a hierarchical Bayesian framework.

We evaluated the progress towards the targeted reduction of VL incidence in the years 2013–2023, by considering the incidence in the year 2013 as the reference rate for the respective block. This will make it possible to compare incidence rates in subsequent years. The expected counts therefore represent the total number of VL cases that one would expect if the population in block *i* had the same *per capita* as in 2013.

**2.4.3. Model description.** We considered non-parametric models with different space-time interactions [36] to study the risks of VL incidence by incorporating fixed effects (covariates), main spatial, and temporal random effects, and space-time interactions. The spatiotemporal models are flexible to assess the effects of covariates, to describe spatial relationships between blocks, to capture temporal trends that may be or may not be linear and to account for the block-specific characteristics.

As the reported VL cases are variable over time within blocks, we assumed the negative binomial probability distribution to describe the number of reported cases $O_{ijt}$ in block *i*, month *j* and year *t* with mean $\mu_{ijt}$ and a constant dispersion parameter *k*:

$$O_{ijt} \sim Negbin(\mu_{ijt}, k)$$

and

$$log(\mu_{ijt}) = log(E_{ijt}) + log(\theta_{ijt})$$

Where $E_{ijt}$ (expected number of cases in block *i* in month *j* and year *t*) is an offset to control for the population size and $\theta_{ijt}$ is the mean relative risk (RR). The log-relative risk is modelled as

$$log(\theta_{ijt}) = \propto + \sum_{k=1}^{9} \beta_k X_k + (\xi_i + \lambda_i) + (\gamma_{jt} + \varphi_{jt}) + \delta_{ijt} \tag{1}$$

Where α is the intercept or global rate, $\beta_k$ are the 'fixed effects' of the covariates $X_k$, $\xi_i$ and $\lambda_i$ are the structured and unstructured spatial random effects, $\gamma_{jt}$ and $\varphi_{jt}$ are temporally structured, and unstructured random effects capturing the global spatial and temporal patterns associated with unobserved and unknown covariates, and $\delta_{ijt}$ is the spatiotemporal interaction random effect dealing with specific temporal trends in each block or changes in the global spatial pattern with time. The spatiotemporal interaction term concerns the interaction between one of the spatial components (structured or unstructured) with one of the temporal components (structured or unstructured), leading to four types of interactions (Blangiardo and Cameletti, 2015): (i) the simplest is the interaction between the unstructured components of space and time (Type I), (ii) interaction between unstructured spatial component and structured temporal component (the structured temporal component is independent of spatial neighborhoods, Type II), (iii) interaction between structured component of space with unstructured component of time (the spatial structure is independent between the other time points, Type III), and (iv) interaction between structured component of space with structured component of time (temporal dependent structure of each neighbourhood depends on the temporal pattern of adjacent neighborhoods, Type IV).

**2.4.4. Model fitting.** The spatiotemporal model (Eq 1 has been fitted considering one of the priors for space (ICAR, BYM2, LCAR) and (RW1 or RW2), and four types of interaction. For comparison, we also fitted the additive models, i.e., Eq (1) without spatiotemporal

interaction random effects for the ICAR, and BYM2 and LCAR priors.

$$log(\theta_{ijt}) = \propto + \sum_{k=1}^{9} \beta_k X_k + (\xi_i + \lambda_i) + (\gamma_{jt} + \varphi_{jt}) \tag{2}$$

Thus altogether, we have fitted a total of 37 models with covariates, which include (i) main spatial and temporal random effects [9 additive models, combination of different priors for the structured spatial effect (LCAR. BYM2 and ICAR), and structured temporal effect (iid, RW1 and RW2)], (ii) spatiotemporal interactions (27 models) and (iii) a fixed effect. The 27-interaction models include nine type I interaction models (3 structured spatial priors x 3 structured temporal priors), and six each of type II-IV interaction models [3 structured spatial priors x 2 structured temporal priors (RW1 or RW2) for each]. All the models were fitted to data covering the period from 2013–2018 (training period).

**2.4.5. Prior distributions used for model fitting.** We used the intrinsic conditionally autoregressive (ICAR, also called as Besag model and IGMRF, the intrinsic Gaussian Markov random field model) [37], or Besag-York-Mollie (BYM2) [38] or Leroux conditionally auto-regressive (LCAR) [39] priors for describing the spatial random effects. The ICAR prior models the effect of structured ($\xi_i$) and unstructured components together ($\lambda_i$), which cannot be identified independently [40]. Whereas, the BYM2 or LCAR priors, models both spatially structured ($\xi_i$) and unstructured ($\lambda_i$) random effects, independently.

We used the 'besag', 'bym2' and the 'generic1' models to implement respectively the ICAR, BYM2 and LCAR prior distributions available in R-INLA package[41] and implemented in 'R' software. The 'generic0' model, available in R-INLA, was used to describe the structured spatiotemporal interaction effect ($\delta_{ijt}$). The 'rw1' and 'rw2' models available in R-INLA were used to define first (RW1) and second (RW2) order random walk priors for the temporally structured random effect ($\gamma_{jt}$). In all the models, an independent and identically distributed *(i.i.d)* Gaussian prior was assigned to temporally unstructured random effect ($\varphi_{jt}$).

All the models were fitted within a Bayesian framework using the INLA developed by Rue et al. [41] and implemented in the R-INLA package 23.11.26 built in 2023-11-26. (www.r-inla.org). INLA algorithm [41] is a deterministic approach to approximate Bayesian inference for latent Gaussian models (LGMs), including the Bayesian generalized linear mixed models (GLMM) [42]. INLA is both faster and more accurate than MCMC alternatives for LGMs and can be used for quick and reliable Bayesian inference in practical applications [41].

In the Bayesian framework, we need to specify prior distributions for the fixed parameters and the hyperparameters. INLA makes use of a normal diffuse priors with zero mean and a precision equal to 0.001. We have used the default diffuse priors for all the fixed parameters. For random effects, we have used uniform prior distributions on the positive real line for the precisions (LCAR or RW1 or RW2 models) and a standard uniform distribution for the hyperparameters of the random effects (LCAR). For the ICAR and BYM2 models, we have used the default logGamma (1,0.00005) for the log-precisions.

**2.4.6. Model selection.** We selected the best model based on the Deviance Information Criterion (DIC), a Bayesian measure for model performance and complexity [43], and the 'Watanabe-Akaike Information Criterion' (also known as 'widely applicable information criterion ', WAIC [44]. The DIC is the sum of the posterior mean of the deviance $\overline{D}$ (a measure of goodness of fit) and the number of effective parameters $P_D$ (a measure of model complexity), i.e.

$$DIC = \overline{D} + P_D$$

Models with the lowest DIC and WAIC value provide the best trade-off between model fit and complexity.

**2.4.7. Model validation and predictive performance.** The best fitting model was validated with a test data set (2019–2020) and was further used to forecast VL incidence beyond the period of observations (2021–2023). The monthly data on covariates extracted for the period, January 2021 –March 2023 and the data derived for the period from April 2023 to December 2023 were used for model prediction. The VL case counts from June 2021- December 2023 were not available for comparison with model prediction.

We assessed the explanatory performance of the best fitting model by examining how much the model estimated number of cases agree with observations by calculating the Pearson's correlation coefficient between observations and predictions at block level, cumulated over months for the training period (2013–18). A value of one indicates perfect correspondence between the model predictions and observations. The fit of the model was also assessed by calculating the proportion of times (block x months = 518 x 12 months = 6216 block-months,) the 95% Bayesian credible interval (BCI) for the predicted total number of VL cases that could capture the observed VL cases per month in each year.

We also assessed the predictive performance of the best fitting model by calculating the *adjusted* version of the probability integral transform (PIT) histograms for discrete data. The PIT evaluates the statistical consistency between the probabilistic forecast and the observation for the test period [45,46]. An *adjusted* PIT is defined as,

$$PIT_{ijt} = Pr(O_{ijt}^{new} < O_{ijt}|O_{-ijt}) + 0.5 \times Pr(O_{ijt}^{new} = O_{ijt}|O_{-ijt})$$

where, $O_{-ijt}$ being the observation vector with $ijt^{th}$ component omitted. The PIT is the value (lie between 0 and 1) that the cumulative predictive distribution function attains at the observation. Deviations from uniformity in a PIT histogram indicates model deficiencies: U-shaped or inverted-U shaped histograms indicate under or over dispersed predictive distributions.

**2.4.8. Assessing significance of predictors.** The significance of the fixed effects parameter estimates was assessed comparing the posterior mean and 95% BCI. The 95% BCI is interpreted as the interval that covers the true parameter value with a probability of 95%, given the evidence provided by the observed data. The 95% BCI for a fixed effect that does not include 'zero' is considered as a significant predictor.

## 3. Results

### 3.1. VL incidence in Jharkhand and Bihar

Fig 2A shows the temporal evolution of crude VL incidence rates (per 10,000 persons per year) in the states (blue dashed lines) of Jharkhand and Bihar and the blocks (grey) of respective states. The state level incidence rates showed a declining trend over the years from 2013 to 2020 in both states, although the rates are above 1 per 10,000 population until 2020 in Jharkhand, and below 1 after 2015 and continued to decline steadily over time in Bihar. The incidence rates tend to decline in most of the blocks over the years within each state and are highly variable among blocks within the respective states. Clearly most of the blocks present crude rates that are higher than the average for the respective states. The affected blocks of Jharkhand on average (3.2 per 10,000 population-year) have much higher incidence than Bihar (0.59 per 10,000 population-year). Further, the rates are highly variable among blocks in the state of Jharkhand (range:0.0–15.8 per 10,000 population-year) when compared to Bihar (range: 0.0–6.8 per 10,000 population-year).

Fig 3 shows the status of elimination in Jharkhand and Bihar. In Jharkhand, the percentage of blocks below the elimination threshold was nil in 2013, around 20% until 2017 and thereafter started increasing to reach 60% in 2020. In Bihar, the percentage of blocks with incidence below the elimination threshold was 70% in 2013 and >90% in 2020.

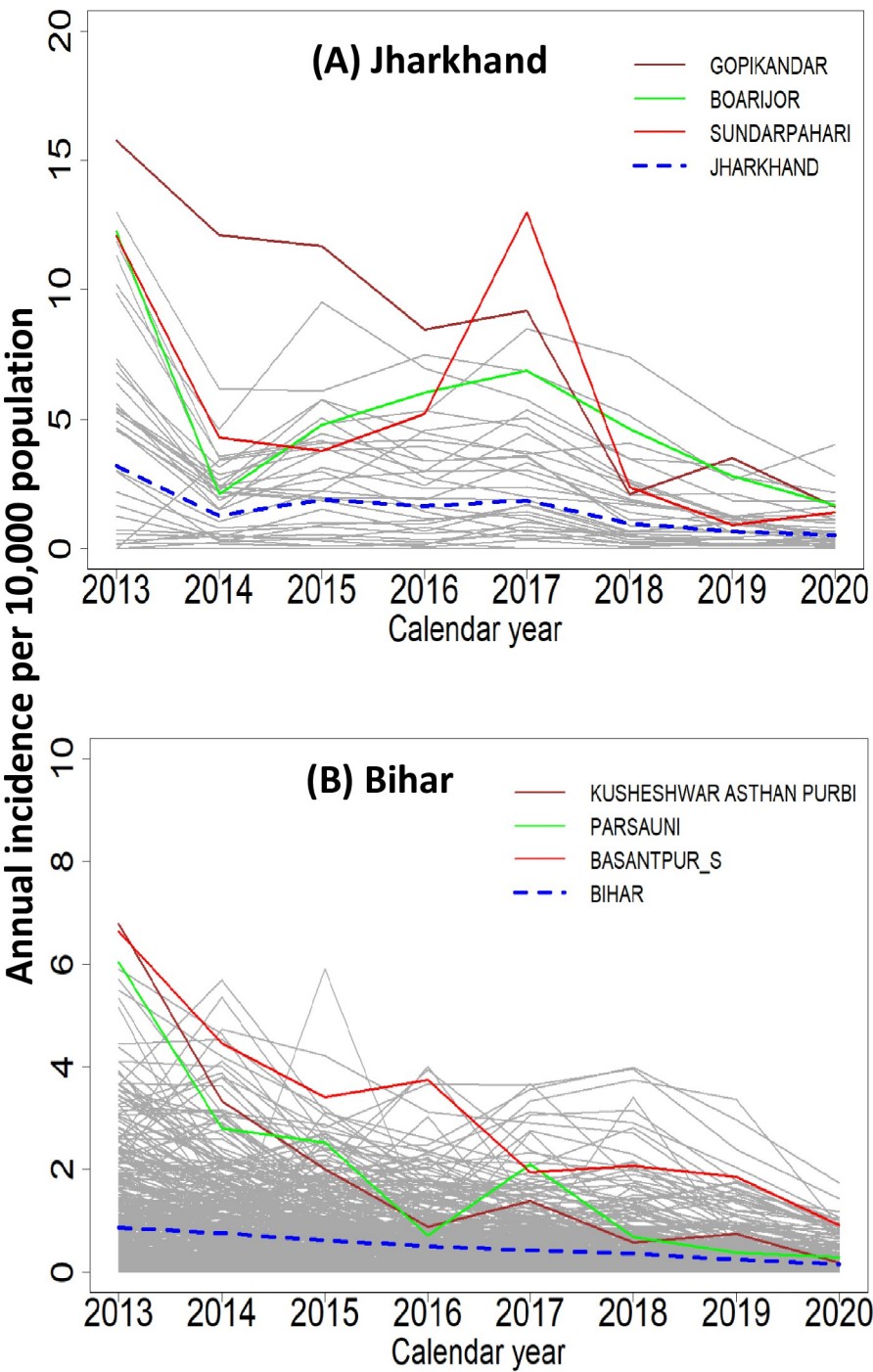

**Fig 2. Evolution of the crude rates of VL incidence in (A) Jharkhand and (B) Bihar in the period 2013–2020.**
Dotted line: state average. Solid grey lines: block wise incidence per year. Thick solid (red, green, and pink) lines:
blocks with > 12 cases (Jharkhand) and > 6 cases (Bihar) per 10,000 population in 2013.

## 3.2. Association of VL incidence with covariates

The spatiotemporal patterns of environment, climatic and bioclimatic variables, and spatial
pattern of population density are depicted in S1 Fig. Fig 4 shows the spatial and temporal

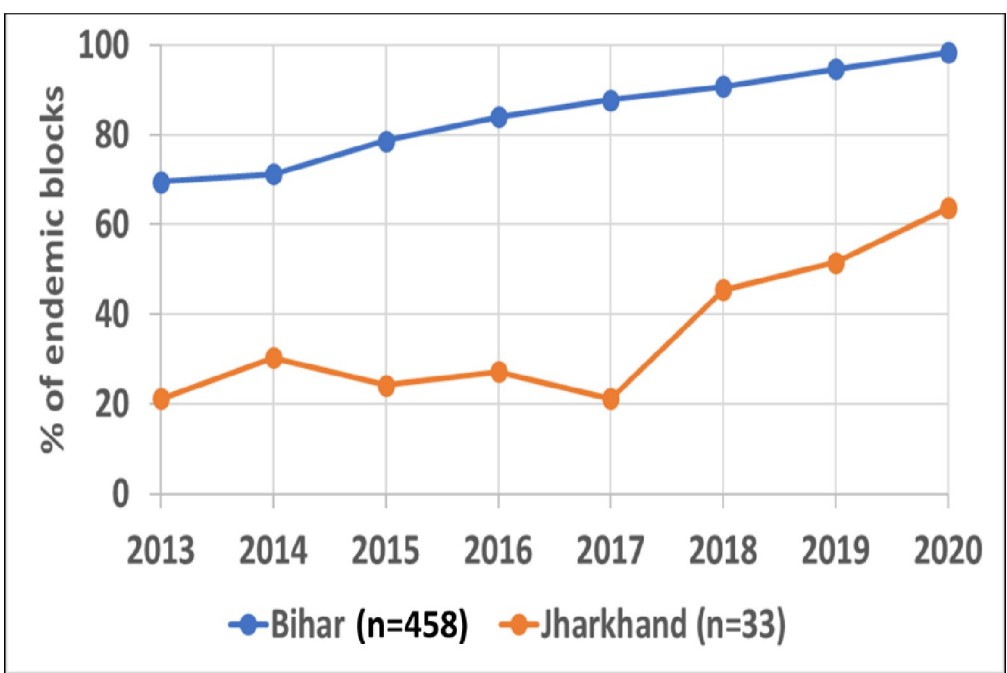

**Fig 3. Percentage of endemic blocks in Bihar (blue) and Jharkhand (red) with incidence below the elimination threshold (<1 per 10,000 population) in each year.**

patterns of correlations of crude VL incidence with each covariate. The spatial correlations range between -0.7 and 0.8 for all the covariates. Conservative assessment showed that correlations of VL incidence with monthly temperature ($X_1$), isothermality ($X_2$), precipitation ($X_3$), maximum temperature ($X_4$), minimum temperature ($X_5$), soil moisture ($X_6$), population

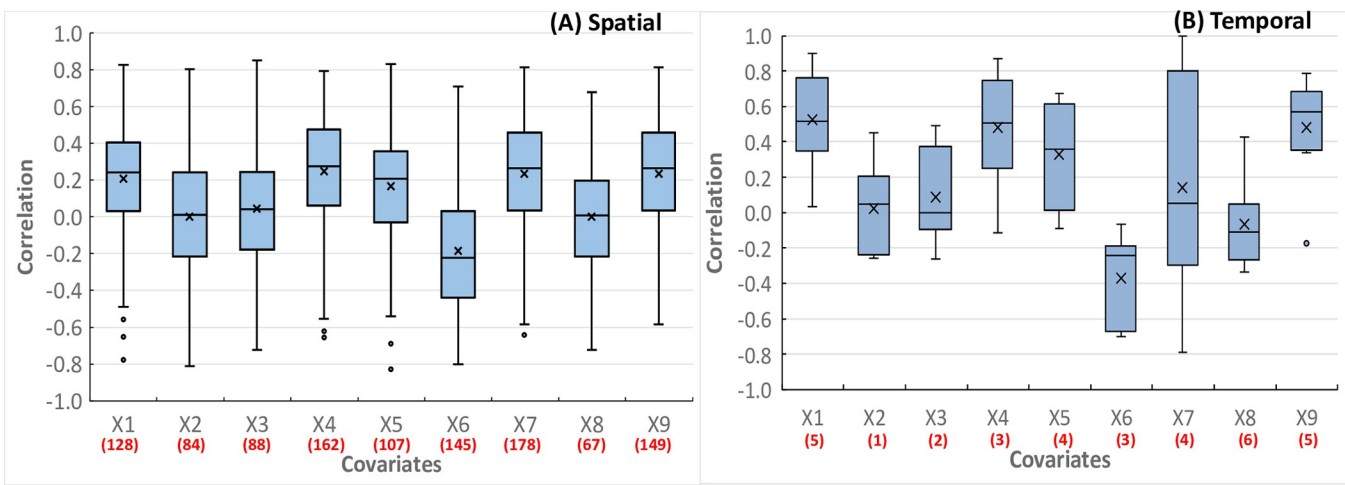

**Fig 4. Boxplots showing the (A) spatial (518 blocks) and (B) temporal (2013–2020) patterns of correlations between VL incidence with each covariate:** monthly temperature ($X_1$), isothermality ($X_2$), precipitation ($X_3$), maximum temperature ($X_4$), minimum temperature ($X_5$), soil moisture ($X_6$), population density ($X_7$), enhanced vegetation index ($X_8$) and land surface temperature ($X_9$). The boxes show the 25th and 75th percentiles of the distribution of the correlation values, the horizontal line across the box is the median and the 'x' is the mean. The whiskers extend to 1.5 times the height of the box (i.e. the interquartile range, IQR). If the data are distributed normally, approximately 95% of the data are expected to lie between the inner fences. Values more than 1.5 IQR's but less than 3 IQR's from the end of the box are labelled as outliers (o). The values in parentheses below each covariate show the number of blocks or years in which the correlations are significantly different from zero at P<0.2.

density ($X_7$), enhanced vegetation index ($X_8$) and land surface temperature ($X_9$) are significantly different from zero (P<0.2) in at least 67 of the 518 blocks (Fig 4A). The correlations of VL incidence with each covariate is highly variable between blocks and its magnitude is in the same or opposite direction depending on the covariate.

Fig 4B shows the temporal correlations of VL incidence with each covariate. The correlations range between 0.4 and 0.9, -0.26 and 0.45, -0.26 and 0.49, -0.11 and 0.87, -0.09 and 0.67, -0.7 and 0.07, -0.79 and 0.98, -0.34 and 0.43, and -0.17 and 0.78 for $X_1$, $X_2$, $X_3$, $X_4$, $X_5$, $X_6$, $X_7$, $X_8$, and $X_9$ respectively. Conservative assessment showed that correlations of VL incidence with each covariate are significantly different from zero (P<0.2) in at least one of the 8 years (2013–2020) (Fig 4B). The temporal patterns of correlations indicate that the association of VL incidence with each covariate is relatively less variable over years compared to spatial variability.

Since all the nine covariates were significantly associated with incidence conservatively at P<0.2, at least in one of the 518 blocks or years (2013–2020), we included all of them in the subsequent Bayesian spatiotemporal modelling of the VL incidence.

## 3.3. Model comparison and performance

The model selection criteria (DIC, and WAIC) for the complete set of models are displayed in S2 Table. A comparison of DIC and WAIC for the 37 models showed that all additive and interaction models provided better fit than a fixed effect model (both DIC and WAIC > 20,000 points). All the random effects models with RW1 prior for time provided better fit than models with RW2. The DIC and WAIC values were remarkably lower for Type II and IV interaction models than Type I and Type III models with RW1 prior for time, irrespective of the priors used for structured spatial component (DIC or WAIC > 2500 points for Type I and > 1900 points for Type III than for Type II or Type IV). Further, comparison of Type II and Type IV interaction models with RW1 prior showed that Type IV interaction models with covariates with any of the priors for space (LCAR, BYM2, and ICAR) was found to perform better than Type II interaction models (at least 300 points less for Type IV interaction model). Among the priors for space in combination with RW1 prior for time, Type IV interaction models with LCAR, BYM2, and ICAR for space in combination with RW1 prior for time (Models: 32, 34, and 36; S2 Table) provided better fit than their combination with RW2 prior: DIC about 3000 points lower for type IV with LCAR, BYM2, and ICAR priors for space and RW1 for time. Among the three models, models 34 and 36 are relatively comparable (difference in DIC: 16) and better than model 32 (DIC > 130 points).

S2 Fig displays the PIT (probability integral transform) histograms for the models with fixed effects (worse fit, Model No. 1), and type IV interaction models (32–37). The type IV interaction models include the significant covariates and LCAR or BYM2 or ICAR prior for space and RW1 or RW2 prior for time. Clearly, the forecasts by type IV interaction model with BYM2 or ICAR and RW1 priors, are reasonably consistent with observations (PIT histograms approximately uniform) compared to that by type IV interaction models with BYM2 or LCAR and RW2, and LCAR and RW1 priors for space and time and covariates. Further, the PIT histograms demonstrate that the type IV interaction model (with BYM2 or ICAR and RW1) with the addition of environmental, climatic and bioclimatic covariates led to a better fit of the model to the data when compared to the fixed effect model. Although, the BYM2 and ICAR models with RW1 prior are comparable, the BYM2 can be more useful to provide independent estimates for the structured and unstructured random effects than the ICAR model. Therefore, in all subsequent sections we have presented the results based on type IV interaction model with BYM2 prior for space and RW1 prior for time with covariates.

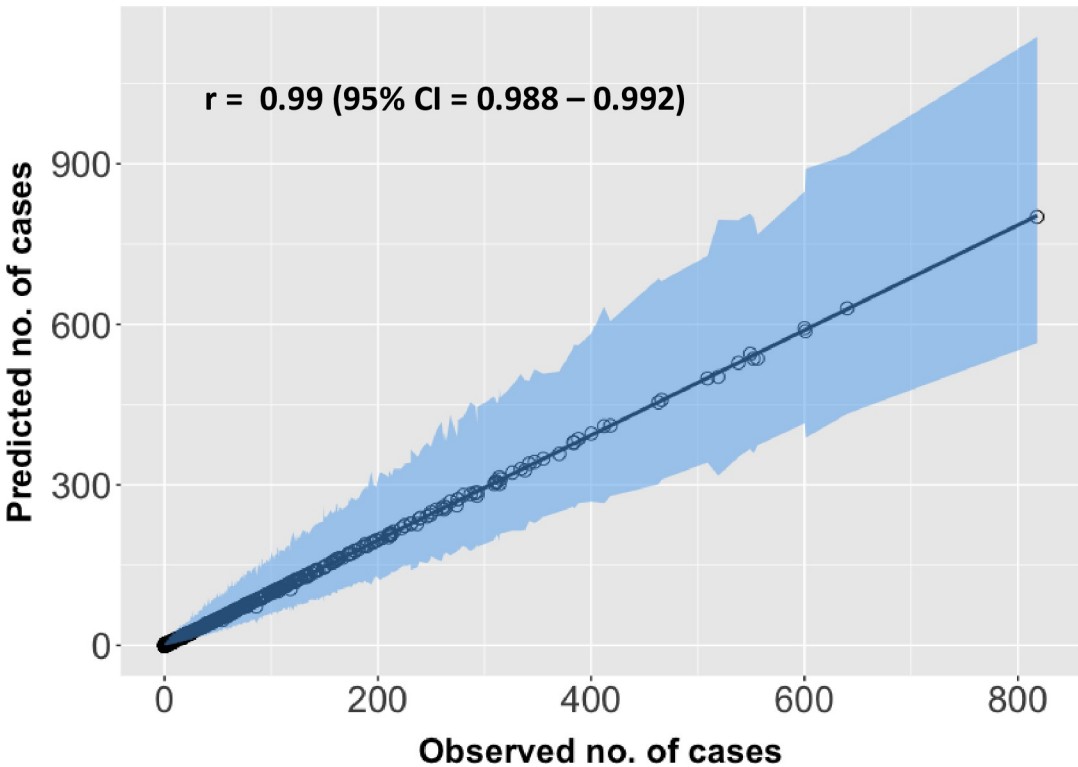

**Fig 5. Comparison of observed the and predicted cases in each block, showing the model's goodness of fit.** The solid diagonal line indicates where values should lie for a perfect correlation between predictions and observations. The predicted number of cases for each block is an aggregate of cases for all the months during the training period, 2013–2018.

Fig 5 compares the model predictions for the entire training period (2013–2018) with observed total VL cases reported for each block. The Pearson correlation coefficient was 0.99 (95% CI: 0.988–0.992, P<0.0001) indicating a high degree of correspondence between model predictions and observations over block.

Fig 6 compares the predictive power of models with only fixed effect, and type IV interaction model (BYM2 prior for space and RW1 for time) with covariates for the training (2013–2018) and test (2019–2020) periods. The predictive powers increased with calendar year for the type IV interaction model but for the fixed effect model the increase was at a lower level: 53 vs 91% in 2013 to 50 vs 95% in 2018 (training period), and 57 vs 98% in 2019 and 39 vs 99.9% in 2020 (test period, Fig 6). The overall predictive power was 52.0% for fixed effect model and 94.3% for type IV interaction model.

### 3.4. Predictors of VL incidence

Fitting the fixed effect model showed that VL incidence is positively associated with minimum temperature, enhanced vegetation index (EVI), isothermality, and precipitation, and negatively with monthly mean temperature, maximum temperature, land surface temperature (LST), soil moisture and population density (Table 1, 95% credible intervals do not include zero). However, the best fitting model (Type IV interaction with BYM2 for space and RW1 for time with fixed effects) showed that LST and population density were the only two variables negatively associated with VL incidence (Table 1, 95% credible interval does not include zero). Consequently, the blocks with low LST or population density tend to have high incidence of VL and vice versa.

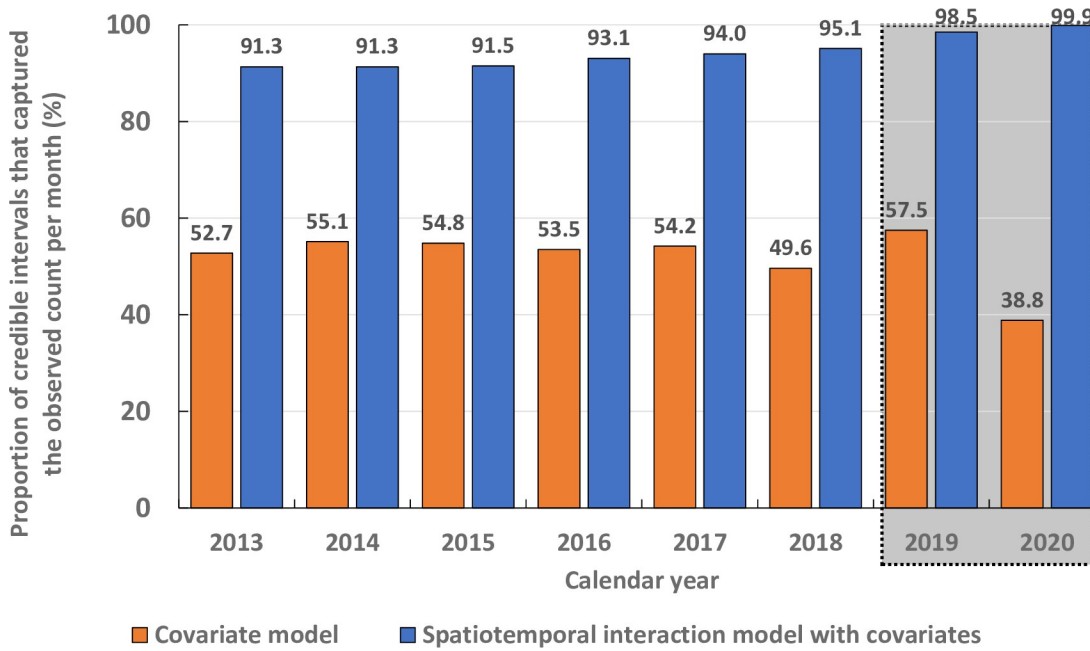

**Fig 6. Model predictive power, calculated as the proportion of 95% credible intervals that could capture the true VL cases per month in each calendar year during training (2013–2018) and test (2019–2020, inside the shaded rectangle box) periods.**

## 3.5. Overall spatial risk pattern and exceedance probability

Fig 7 maps the block-specific relative risks (Fig 7A) and the probabilities that these block-specific risks are greater than 1, (Fig 7B). After adjusting for the fixed covariate effects and time random effects, the residual spatial risk pattern provides evidence for a strong spatial heterogeneity. Many of the blocks from east to west in the central and northern part of Bihar and all the

**Table 1. Posterior mean, SD, and 95% credible intervals for the fixed effects based on the model with covariates alone and the best fitting type IV spatiotemporal interaction model with covariates.**

| Covariates | Model with covariates only | | | | Type IV interaction model with covariates | | | |
|---|---|---|---|---|---|---|---|---|
| | Posterior mean | Standard deviation (SD) | 95% credible interval | | Posterior mean | Standard deviation (SD) | 95% credible interval | |
| | | | Lower limit | Upper limit | | | Lower limit | Upper limit |
| Intercept | -0.379* | 0.011 | -0.400 | -0.357 | -1.339* | 0.352 | -2.028 | -0.650 |
| Monthly mean temperature (BIO 1, $X_1$) | -0.633* | 0.096 | -0.821 | -0.444 | -0.270 | 0.357 | -0.970 | 0.431 |
| Isothermality (BIO 3, $X_2$) | 0.900* | 0.044 | 0.813 | 0.987 | -0.079 | 0.072 | -0.221 | 0.063 |
| Precipitation (BIO 12, $X_3$) | 0.087* | 0.019 | 0.051 | 0.124 | 0.038 | 0.026 | -0.014 | 0.090 |
| Maximum temperature ($X_4$) | -0.527* | 0.082 | -0.688 | -0.366 | 0.319 | 0.192 | -0.057 | 0.695 |
| Minimum temperature ($X_5$) | 1.799* | 0.109 | 1.586 | 2.012 | 0.063 | 0.256 | -0.437 | 0.564 |
| Soil moisture ($X_6$) | -0.057* | 0.017 | -0.090 | -0.024 | -0.048 | 0.028 | -0.102 | 0.006 |
| Population density ($X_7$) | -0.270* | 0.011 | -0.291 | -0.248 | -0.241* | 0.054 | -0.348 | -0.135 |
| Enhanced vegetation index ($X_8$) | 0.105* | 0.012 | 0.081 | 0.128 | -0.002 | 0.010 | -0.021 | 0.017 |
| Land surface temperature ($X_9$) | -0.129* | 0.017 | -0.163 | -0.095 | -0.041* | 0.020 | -0.081 | -0.001 |

* Significant variables, 95% credible interval does not include zero

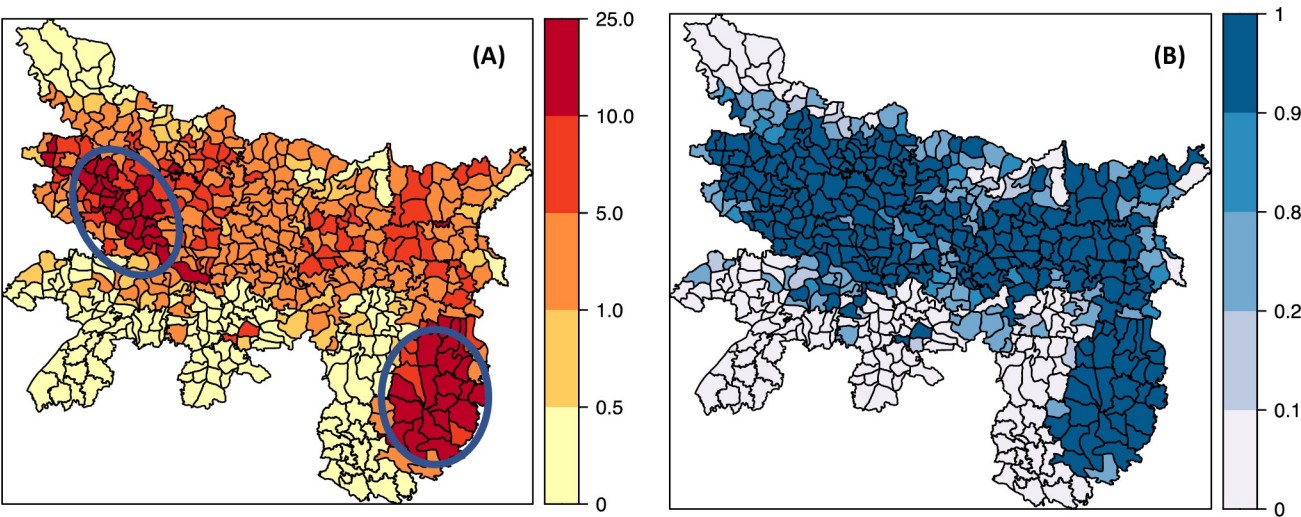

**Fig 7. Posterior means of the block-specific (A) relative risks $\zeta_i = exp(\xi_i)$ and (B) posterior probabilities $P(\zeta_i>1/O)$ that the relative risks are greater than 1.** Relative risk is >5.0 for blocks lie within circles. Probability that relative risk exceeds 1.0 is >90% for most of the blocks and is across the centre of Bihar and almost all the endemic blocks in Jharkhand. The risk in non-endemic blocks in both Bihar and Jharkhand are less than 0.5 and the probability that the risk greater than 1 is negligible (P<0.1). Block level shapefile for Bihar and Jharkhand were developed in ArcGIS software (https://www.arcgis.com) by digitization tool using base layer from the India village directory, Census of India 2011, download from https://lgdirectory.gov.in.

endemic blocks in northern part of Jharkhand present a greater risk (relative risk >5.0) than the blocks in the north-east and southern part of Bihar (relative risk <0.5).

The risk in non-endemic blocks in both Bihar and Jharkhand are less than 0.5 (Fig 7A) and the probability that the risk greater than 1 is negligible (P<0.1) (Fig 7B). The probability risk map could be divided into three groups: (i) blocks whose probabilities are greater than 0.9, (ii) blocks with probability of risk falling between 0.8 and 0.9 and (iii) blocks with probabilities smaller than 0.2. The blocks with probabilities over 0.9 are classified as high-risk blocks.

## 3.6. Overall temporal trend in relative risk

Fig 8 shows the temporal relative risk of VL (all blocks combined for a month) in Bihar (Fig 8A) and Jharkhand (Fig 8B) during the training and test periods. The monthly relative risk shows an annual peak in both states. Most of the time the risk fluctuates around one until 2020. Thereafter, it tends to increase above one and is 2 times higher in Jharkhand compared to Bihar. However, the 95% BCI indicates that the risk was above one for the entire study period in both states; In Jharkhand, it is about 7 times higher than that in Bihar (70 vs 10), indicating that some other factors are differently influencing the two states.

## 3.7. Spatiotemporal patterns in annual incidence

Fig 9 compares the observed and predicted spatiotemporal patterns of annual incidence of VL by blocks during the training (2013–2018), testing (2019–2020) and forecasting (2021–2023) periods. The predicted incidences agree with the observed declining trends in most of the blocks during the training period, and in 2019 of the testing periods. In 2020, however, the observed incidence rates were lower than that of predicted. Predictions beyond the period of observations (2021–2023) showed that the incidence is more or less stable over time. The predictions also show that most of the endemic blocks in Jharkhand could have restored back to the 2013 status (incidence above 10 per 10, 000 population) and that the blocks in Bihar are with an incidence range of 1.0–5.0 per 10,000 population per year; the nonendemic

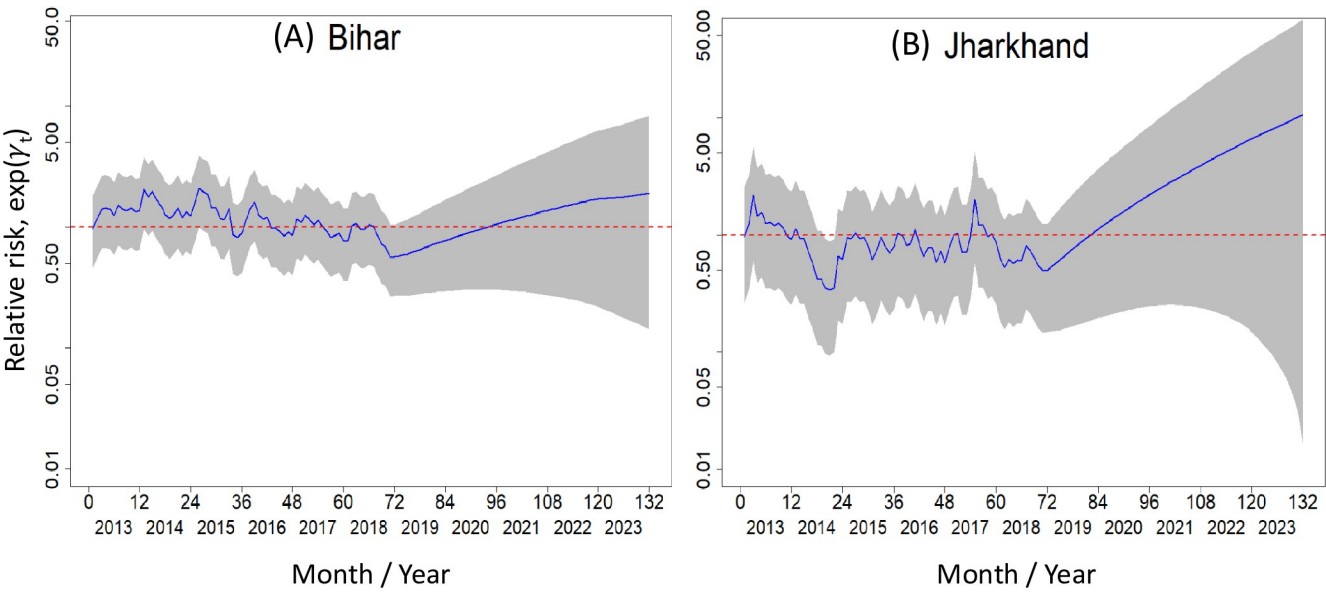

**Fig 8. Overall temporal trends of VL incidence relative risk (blue solid line) in (A) Bihar and (B) Jharkhand, 2013–2023.** Relative risk is 1 on the dotted horizontal line. The grey shaded area is the 95% Bayesian credible interval for the risk.

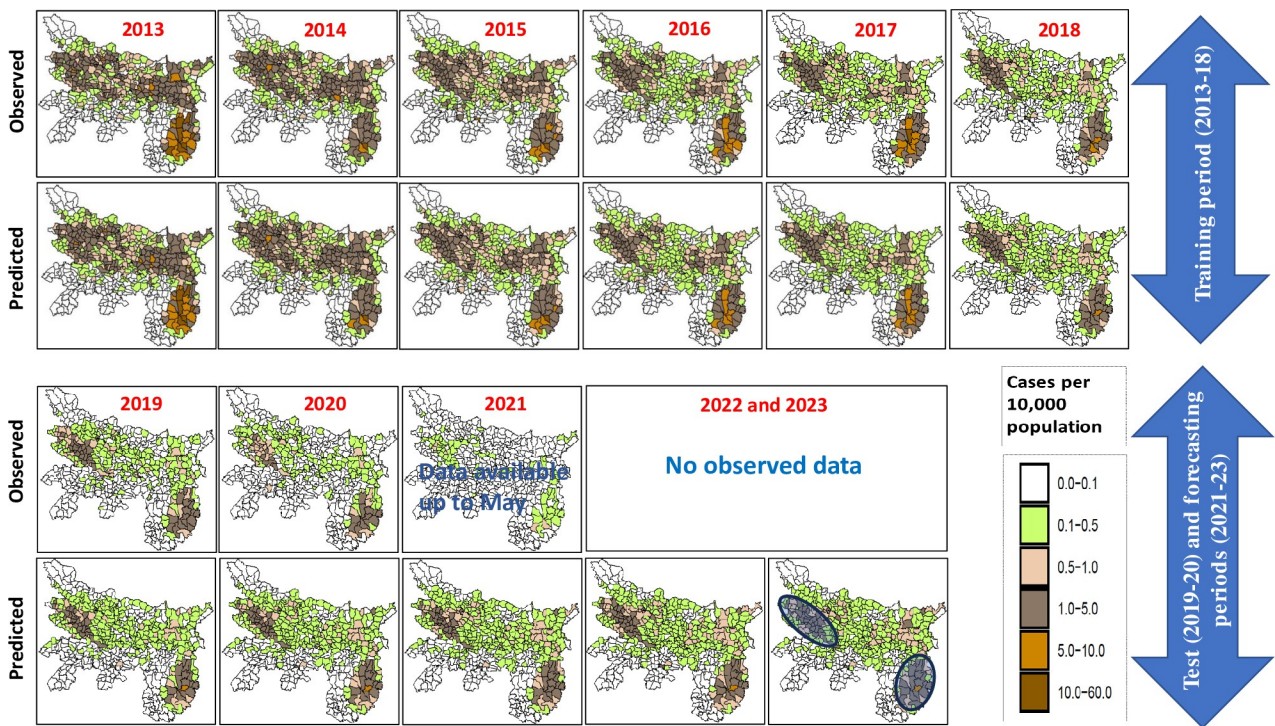

**Fig 9. Spatial pattern in observed and predicted annual VL incidence per 10,000 population during the training (2013–2018), testing (2019–2020) and forecasting (2021–2022) periods in the states of Bihar and Jharkhand.** Model predicted incidence agree with declining trend in most of the blocks during the training, and in 2019 of the testing periods. In 2020, however, the observed incidence rates were lower than that of predicted. Predictions beyond the period of observations (2021–2023) showed that the incidence is more or less stable over time. Areas in circles indicate that the incidence in these blocks are likely to exceed the elimination threshold. Block level shapefile for Bihar and Jharkhand were developed in ArcGIS software (https://www.arcgis.com) by digitization tool using base layer from the India village directory, Census of India 2011, download from https://lgdirectory.gov.in.

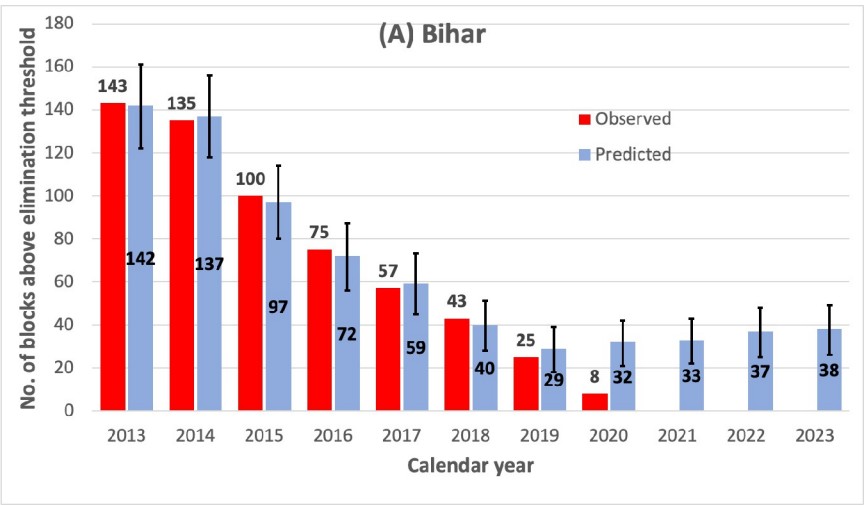

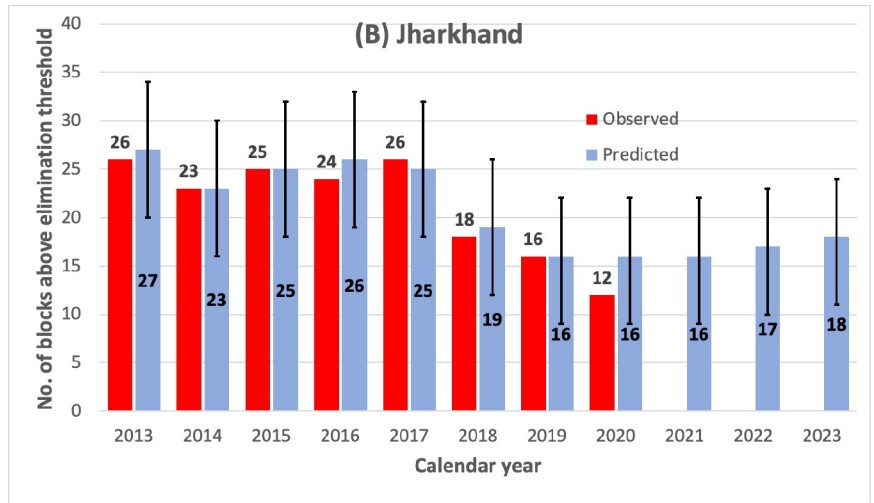

**Fig 10. Observed and predicted trends in number of blocks above elimination threshold in the states of Bihar (A) and Jharkhand (B).** Values above or inside the bars indicate the observed or model predicted number of blocks exceeding elimination threshold in each year. Error bars show the 95% confidence interval for predicted number of blocks above the elimination threshold. The upper 95% CI for 2021–2023 show that the incidence is likely to exceed the elimination threshold in 33, 37 and 38 blocks in Bihar and 16, 17, and 18 blocks in Jharkhand.

neighbouring blocks both in Bihar and Jharkhand, as expected, are well below the elimination threshold (Fig 9).

Fig 10 compares the observed and predicted number of blocks exceeding the elimination threshold in Bihar (Fig 10A) and Jharkhand (Fig 10B) by years. In Bihar, the predicted number of blocks above the elimination threshold are in agreement with observations until 2019 (95% CI includes the observations) but in 2020, the observations are fewer than that model predicted. Predictions for 2020 indicate that 32 blocks could have exceeded the elimination threshold against 8 blocks that were actually reported (Fig 10A). The predictions for 2021–2023 also indicate that 33–38 blocks could have exceeded the threshold in Bihar.

In Jharkhand, the number of blocks above the elimination threshold are consistent with observations (95% CI includes observation) during the training and testing periods (Fig 10B).

Predictions beyond the period of observations (2021–2023) showed that numbers of blocks likely to exceed elimination threshold ranged from 16–18.

A list of blocks in which the annual incidence above the elimination threshold is given in S3 Table.

### 3.8. Model performance in selected blocks

We have compared the best fitting model performance in two different epidemiological settings: (i) blocks where the observed incidence in 2013 is well below the elimination threshold and (ii) in those where it is above the elimination threshold. The results for a few selected blocks from each setting are presented below.

Generally, the model predictions agree with the observed declining trends in many blocks of both the settings during training and testing periods. Predictions beyond the period of observations (2021–2023) showed that the annual incidence is more likely to exceed the elimination threshold in the blocks where the reported VL incidence was > 6 per 10,000 population in 2013. For example, in the three blocks each from Bihar (Kusheswar Asthan Purbi, Pursani and Basantpur_S) and Jharkhand (Gopikandar, Boarijor and Sundarpahari), where the annual incidence was above 6 in 2013, the model predictions agree with the exponentially declining observed trends in incidence (Fig 11) in all the blocks until 2020. However, the predictions for 2021–2023 indicate that VL incidence is likely to be above (one in Bihar and all the three blocks in Jharkhand) the elimination threshold.

Fig 12 illustrates the model performance in (A) two of the blocks each in Bihar (Barauni and Gopalpur) and Jharkhand (Mandro and Sahibganj) where the annual incidence in 2013 was '0', and (B) in another two blocks each in Bihar (Balia and Sahepur Kamal) and Jharkhand (Ranishwar and Barharva) well below the elimination threshold (0.1 to < 1 per 10,000

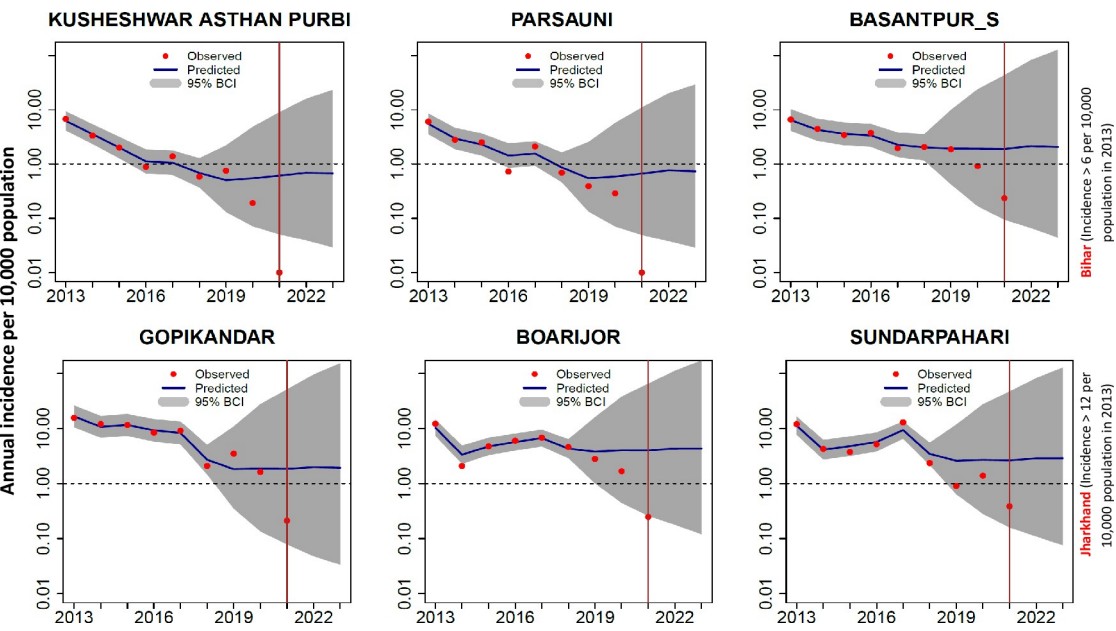

**Fig 11. Spatiotemporal variation in 3-blocks each with annual incidence > 6 or > 12 per 10,000 population in 2013 respectively in Bihar (top panel) and Jharkhand (bottom panel).** The predicted trends closely mimicked the declining observed trends in all the 3 blocks for both Bihar and Jharkhand until 2020. The red vertical line indicates the period up to which VL case data are available (up to May 2021). Predictions for 2021–2023 indicates the likelihood of exceeding the elimination threshold (1/ 10,000 population) in one block Bihar and all the three selected blocks in Jharkhand.

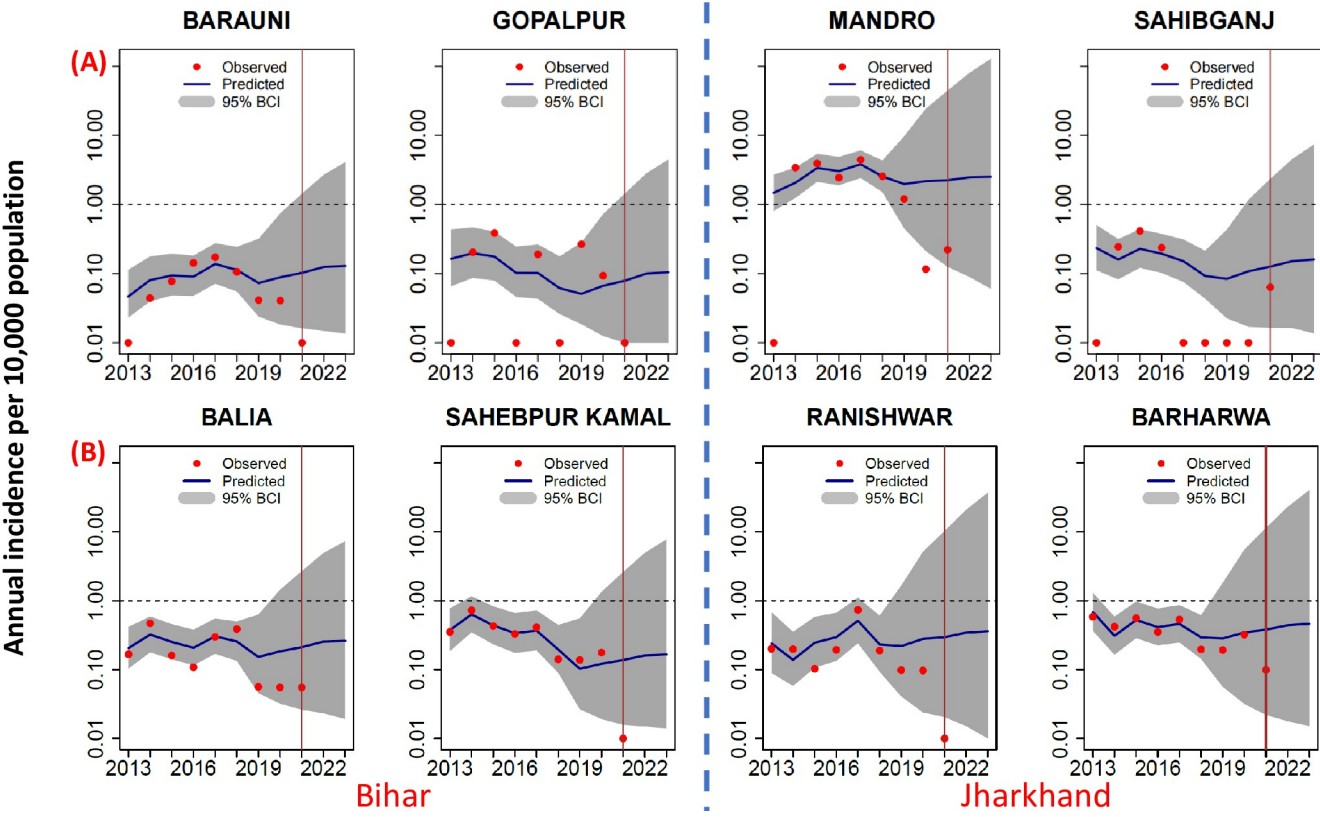

**Fig 12. Model performance in selected blocks of Bihar and Jharkhand where the annual incidence was (A) '0', and (B) 0.1–1.0 per 10,000 population in 2013.** Predicted trends closely mimicked the declining observed trends in all the 8 blocks, and sustained below the elimination threshold (1/10, 000 population) in all the blocks. A value of '0.01' was added to blocks in which the incidence was '0' to plot in the log scale.

population).The observed incidence continued to be below the elimination threshold during training, testing and forecasting periods in both settings. The model predictions for all the blocks are in line with observed trends but stochastic effects dominate the patterns in the blocks with '0' incidence during 2013: high variability in the observed and predicted trends (Fig 12B).

## 4. Discussion

In this study, we have developed and validated a spatiotemporal modelling framework for predicting block level VL incidence for all the endemic blocks (491) and a few selected non-endemic blocks (27) in Bihar and Jharkhand states. Earlier, Nightingale et al. [32] applied a statistical model on surveillance data (2013–2018) collected from the same study area and forecasted monthly VL incidence at the block level, which predicted the trends 1-month, 3-months and 4-months ahead. The forecasts could be used to help the programme for logistics management in advance. The authors considered that the current incidence in a block is related to (i) previous incidence in the same block (autoregressive effect), (ii) previous incidence in surrounding blocks (neighbourhood effect), and (iii) intrinsic block level factors (geographical or demographic), called as, 'endemic '. The model considered population density as the only 'endemic' factor. In this study, we extended the work of Nightingale et al.[32] to improve the predictive power of the model by (i) incorporating data on environmental, climatic, bioclimatic and demographic factors that influence VL transmission dynamics, (ii)

adding more data on monthly reported VL cases (until May 2021), and (iii) fitting spatial, temporal and spatiotemporal random effects models to minimize the variability unexplained by the fixed effect covariates using the computationally less demanding INLA approach [18, 41, 47] by means of the R-INLA package (www.r-inla.org). The best fitting spatiotemporal model (type IV interaction model with covariates, no. 34 in S2 Table) could better explain the observed spatial and temporal patterns of annual VL incidence in most of the blocks (99% of observations over 24 months of testing period, 2019–2020, Fig 6), compared to the model of our earlier study (94% of observations over 24 months testing period, 2017–2018) [32]. The DIC values for the model with fixed effect has declined from 103,428.99 to 75,159.58 (27% reduction) for the type IV interaction model with covariates. Further, the overall predictive power for the years 2013–2020 was relatively higher for type IV interaction model than the model with fixed effects only (94.3 vs 52.0%) (Fig 6). Thus, we have shown that incorporation of environmental, climatic and demographic covariates into a spatiotemporal interaction model improved its predictive ability highlighting the importance of these data in driving the spatial and temporal heterogeneity in VL incidence.

The predictions using the best fitting model could also describe the observed trends in two heterogeneous settings, i.e. in the blocks with VL incidence above elimination threshold in 2013 and in the blocks with incidence well below the threshold, in the two states with widely differing incidence trajectories over both space and time (Figs 11 and 12). Model predictions for non-endemic blocks bordering endemic blocks in both Bihar and Jharkhand indicates that the risk of VL in these blocks continued to be negligible (Fig 7).

Before arriving at the final spatiotemporal model, we have fitted a fixed effects model relating covariates with VL case counts. The model identified significant positive association of VL incidence with minimum temperature, EVI, precipitation and isothermality, and a negative association with monthly mean temperature, maximum temperature, LST, soil moisture and human population density. A systematic review showed that the association of VL and CL incidence with climatic conditions differed among geographical regions [48]. In the Gangetic plain of Bihar, VL incidence is positively associated with environmental (presence of water bodies, woodland and urban, built-up areas, soil type) and climatic (air temperature, relative humidity and annual rainfall) factors [29, 49]. In Muzaffarpur district, Bihar, the VL incidence was positively associated with rainfall and negatively with relative humidity [50]. Studies in Brazil reported an inverse relation of VL incidence with annual or mean of 3-year precipitation, and socioeconomic factors and no association with presence of vegetation [21, 51]. Another study in Brazil found that areas with high human population density and abundant vegetation are associated with high VL incidence [52]. Our study shows that the population density was negatively associated with VL incidence, although there was a positive correlation with EVI. This could be attributed to the fact that a large proportion of population at risk (>75%) live in rural blocks having relatively low population density (Bihar: 1005.4 in rural and 5057.8 per km$^2$ in urban; Jharkhand: 323.4 in rural and 3527.5 per km$^2$ in urban) and abundant vegetation (as indicated by EVI), which are positively associated with VL incidence. Abundance of vegetation seems to make the difference as it can affect the LST and availability of sandfly breeding and resting habitats. Therefore, sandfly density (human-vector contact) and VL transmission are expected to be higher in rural areas, where population density is low, and vegetation is abundant. There were studies that demonstrated either positive or negative association of *Phlebotomus argentipes* (the vector of VL) density with climatic factors [53] including NDVI/vegetation cover. In our model, we could not consider sandfly density as the data collected by a standardized method were not available at block level.

The final spatiotemporal model also showed that VL incidence is negatively associated with LST and population density, as indicated by fixed effect models. However, the final model

showed that the association of VL incidence with all other covariates were insignificant. This contradicts with correlation analyses of fixed effect model that showed significant positive or negative correlations with all the covariates. Generally, in the space-time interaction models, the associations of covariates with response variable are reported to be masked due to inclusion of temporarily correlated random effects and space-time interaction effects (spatially and temporarily correlated) [54]. Such confounding effects could have been alleviated by applying spatiotemporal models with restricted regression or orthogonality constraints approaches [54]. These authors, applying the above techniques, estimated association between dowry deaths and socio-demographic covariates in the districts of UP, India and demonstrated that both restricted regression and orthogonal constraints alleviate confounding. They provided estimates and their standard errors similar to that of the fixed effects model. If the relative risks / predicting response are of primary interest, ignoring confounding is not a problem as the estimates of classical spatiotemporal interaction model (type IV interaction model) are not affected by confounding. Therefore, we have used the results of fixed effects model for inferring associations, and the classical type IV interaction model with covariates (i.e. the final model, No. 34, S2 Table) for all predictions.

Model predictions beyond the period of observations (2021–2023) showed that the annual incidence is likely to exceed the elimination threshold in the blocks where the reported VL incidence was > 6 per 10,000 population in 2013. More than 54% (18/33 blocks) and 8% (38/458 blocks) of the blocks are likely to exceed the elimination threshold in Jharkhand and Bihar respectively in 2023 (S3 Table). The results are to be compared with observed VL incidence in these blocks in 2021–2023. The block level data for this period were not available until completion of the study for such comparison. However, the combined data available on VL incidence for 2021–2022 for all the four VL endemic states in India indicate a significant decline in VL burden (1276 cases in 2021, 810 cases in 2022). In Bihar alone, the number of cases declined from 893 in 2021 to 547 in 2022, and in Jharkhand the corresponding figures are 275 and 187 for the same period [5]. Although the national KAELP had integrated the VL case detection and treatment with the COVID activities [1], it is likely that the cases could have been under reported during this period and many VL cases might have been left undetected. VL cases might have been missed as the early clinical symptoms (fever and asthenia) and laboratory parameters (e.g., leukopenia, thrombocytopenia, and elevated transaminases) might be similar in patients infected with SARS-CoV-2 and those with VL [55]. This might be an important issue, which could result in resurgence, and affect the precision of model predictions, as the model did not account for detection and treatment in the context of COVID-19 pandemic. Therefore, it is likely that the actual number of blocks above the elimination threshold could be more than that is being reported.

The predicted 95% credible intervals were wider for the testing and forecasting periods than that for the training period in a few of the selected blocks (Figs 11 and 12). This suggests that, during this period, the block level incidence was highly variable both spatially and temporally, particularly in blocks with zero incidence prior to intensive intervention in 2013. In our model, we have considered a common dispersion parameter for all the blocks and periods to describe the heterogeneity in incidence. Block-specific dispersion parameters could have accounted for this highly variable incidence. However, as reported in our earlier publication, a block-specific dispersion parameter was not a viable option as some blocks demonstrated zero incidence for prolonged periods with sporadic cases and if used will lead to unrealistically high predictions [32]. These patterns (cases become sparser in space and time) could partially be due to differential intervention efforts, and also expected to be more common as elimination is approached, pointing to the need for focused surveillance, and interventions at village level to stop resurgence and sustain elimination.

Although our model predictions agree with observations during training and testing periods, the study has a few limitations. In our model, we did not consider the role of post kala-azar dermal leishmaniasis (PKDL) cases on VL transmission. Recent xenodiagnoses studies have indicated the role of PKDL cases on the transmission of VL during the inter-epidemic period [56, 57]. Modelling PKDL cases in the endemic blocks along with covariates, including sandfly density could provide more insight on the incidence of VL cases, especially when the incidence in the blocks is maintained below the elimination threshold.

Secondly, it is likely that other co-variate datasets (e.g., human development index, age, and gender of humans, sandfly density, and vector control efforts such as insecticide spray coverage) not considered in this study could have influenced the VL incidence at block level. At the micro level, influence of literacy level, economic status (poverty) and occupations on VL incidence has been reported in India [58–62] and Nepal [58–62]. Inclusion of these covariates in the model might add additional power to forecast incidence.

Third, our model-based analysis and the predictions solely depend on the reported VL cases in different blocks in two endemic states over a time span of 8 years. The quality of surveillance, diagnostic accuracy due to changes in the surveillance activities from only VL to VL with other comorbidities (VL and HIV or COVID-19) might have varied over time. For example, in east Africa, it has been reported that rapid diagnostic test (RDT) is less sensitive in human immunodeficiency virus (HIV) positive patients than in HIV-negative patients [63], whereas, in Indian subcontinent, its accuracy in co-infected patients is yet to be established [64]. As discussed earlier, the diagnostic accuracy is expected to be further lowered during COVID-19 outbreaks as many VL cases might have been undetected / missed due to coinfection of VL and COVID. Therefore, our predictions, particularly during COVID-19 outbreak in 2020 showing higher incidence of VL than that was reported in the two states has to be viewed in this context.

The final limitation is related to fitting type IV interaction models with restricted regression or orthogonality constraints approaches to alleviate the effects of confounding the association of covariates with VL incidence. We could not apply either of these two approaches due to computational constraints to carry out simulations with large dimension of data (518 blocks x 10 years x 12 months per year = 61440 records) used in this analysis. However, as indicated earlier, the above refined approaches with type IV interaction model are required only if one wants to assess the association of covariates with response variable, and predict/forecast from the same model, else if only prediction is required, the classical type IV interaction model suffice. Accordingly, we measured the association of the covariates with VL incidence based on fixed effects model, and the classical type IV interaction model for predicting, and forecasting VL incidence at block level.

Despite the limitations, our work provides a framework that could also be applied to other settings with anthroponotic leishmaniasis to monitor and forecast the incidence of VL. The data for other endemic states (West Bengal and Uttar Pradesh) can also be used to inform the model, improve and update or validate model predictions. Our modelling work demonstrated the usefulness of KAMIS data for predicting trends in settings with non-zero and zero cases. If KAMIS database is integrated with the modelling framework with a dashboard facility, it can be used to predict / forecast VL outbreak or resurgence especially for post elimination periods. The model predictions could also be used for the non-endemic blocks bordering the endemic blocks, which could aid the programme to expand the surveillance and control operations in advance so that the KAEP meet the target of achieving elimination by 2030. This, however, would require collaborative efforts of programme authorities, modelling experts and a good programmer to make it a reality in future.

## 5. Conclusion

Our spatiotemporal modelling framework after accounting for environmental, bioclimatic and demographic factors could better explain the observed spatiotemporal patterns in VL incidence at block level (subdistrict) than the model without covariates. Model predictions agree with >93 and 99% of the monthly-observations for the periods for the training and testing periods. Forecasting beyond the period of observations (2021–2023) indicated exceedance of elimination threshold in 16–18 and 33–38 historically high endemic blocks of Jharkhand and Bihar, although the reported cases for the two states are declining over time. This highlights the need for the programme to keep vigilance and target control measures in the blocks. The model can also be used to monitor risk of resurgence / recrudescence in the blocks where the incidence is well below the elimination threshold, and in the non-endemic blocks bordering the endemic blocks. Model predictions for blocks in which incidence is just above or below the elimination threshold and showing high variability indicate need for more targeted actions such as intensified surveillance and treatment, and preventive measures focused at village level. The model can be used effectively, if the KAMIS database is integrated with the modelling framework.

## Supporting information

**S1 Fig. Maps of covariates (averaged for the period 2013–2020) used in models.** (A) Monthly mean temperature per month (BIO1, ˚C), (B) Isothermality (BIO3, %), (C) average precipitation per month (BIO12, mm), (D) Monthly maximum temperature (˚C), (E) Monthly minimum temperature (˚C), (F) Soil moisture ($m^3/m^3$), (G) Population density (per $Km^2$), (H) Enhanced vegetation index (spectral index), (I) Land Surface Temperature (˚C). Block level shapefile for Bihar and Jharkhand were developed in ArcGIS software (https://www.arcgis.com) by digitization tool using base layer from the India village directory, Census of India 2011, download from https://lgdirectory.gov.in.
(DOCX)

**S2 Fig. Histograms of Probability integral transform (PIT) values for models with fixed effects, and type IV interaction models with LCAR, BYM2 and ICAR priors for space and RW1 or RW2 priors for time.** U shaped histograms indicate under-dispersed predictive distributions, hump or inverse-U shaped histograms point at overdispersion, and skewed histograms occur when central tendencies are biased. Dashed red lines show the histogram height corresponding to perfect calibration.
(TIF)

**S1 Table. Data source and properties of environment, bioclimatic and demographic variables.**
(DOCX)

**S2 Table. Model selection criteria (DIC, $P_D$ and WAIC) for the complete set of models.**
(DOCX)

**S3 Table. List of blocks with annual VL incidence above the elimination threshold (1/10,000 population) in the states of Bihar and Jharkhand during 2021–2023.**
(DOCX)

## Acknowledgments

We gratefully acknowledge Dr Dhingra, former Director of the National Vector-Borne Disease Control Programme (NVBDCP), Ministry of Health and Family Welfare, Government of India for the support and permission to use the KAMIS-data.

## Author Contributions

**Conceptualization:** Swaminathan Subramanian, Purushothaman Jambulingam.

**Data curation:** Swaminathan Subramanian, Rajendran Uma Maheswari, Gopalakrishnan Prabavathy, Mashroor Ahmad Khan, Balan Brindha, Emily S. Nightingale, Nupur Roy.

**Formal analysis:** Swaminathan Subramanian, Rajendran Uma Maheswari, Gopalakrishnan Prabavathy, Mashroor Ahmad Khan, Balan Brindha.

**Funding acquisition:** Ashwani Kumar, Manju Rahi, Mary M. Cameron, Purushothaman Jambulingam.

**Investigation:** Swaminathan Subramanian.

**Methodology:** Swaminathan Subramanian, Purushothaman Jambulingam.

**Project administration:** Ashwani Kumar, Manju Rahi, Mary M. Cameron, Purushothaman Jambulingam.

**Resources:** Purushothaman Jambulingam.

**Software:** Swaminathan Subramanian, Rajendran Uma Maheswari, Gopalakrishnan Prabavathy, Mashroor Ahmad Khan.

**Supervision:** Purushothaman Jambulingam.

**Validation:** Swaminathan Subramanian, Adinarayanan Srividya.

**Visualization:** Swaminathan Subramanian, Rajendran Uma Maheswari, Gopalakrishnan Prabavathy, Mashroor Ahmad Khan, Balan Brindha, Adinarayanan Srividya.

**Writing – original draft:** Swaminathan Subramanian.

**Writing – review & editing:** Swaminathan Subramanian, Rajendran Uma Maheswari, Gopalakrishnan Prabavathy, Mashroor Ahmad Khan, Balan Brindha, Adinarayanan Srividya, Ashwani Kumar, Manju Rahi, Emily S. Nightingale, Graham F. Medley, Mary M. Cameron, Nupur Roy, Purushothaman Jambulingam.

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
