## [Decision Letter · Decision Letter 0]

2 Jul 2023

Dear Dr. Subramanian,

Thank you very much for submitting your manuscript "Modelling spatiotemporal patterns of visceral leishmaniasis incidence in India using environment, bioclimatic and demographic data, 2013-2021" for consideration at PLOS Neglected Tropical Diseases. As with all papers reviewed by the journal, your manuscript was reviewed by members of the editorial board and by several independent reviewers. In light of the reviews (below this email), we would like to invite the resubmission of a significantly-revised version that takes into account the reviewers' comments. 

We cannot make any decision about publication until we have seen the revised manuscript and your response to the reviewers' comments. Your revised manuscript is also likely to be sent to reviewers for further evaluation.

Sincerely,

Alberto Novaes Ramos Jr

Academic Editor

Charles Jaffe

Section Editor

Reviewer's Responses to Questions

**Key Review Criteria Required for Acceptance?**

**Methods**

-Are the objectives of the study clearly articulated with a clear testable hypothesis stated?

-Is the study design appropriate to address the stated objectives?

-Is the population clearly described and appropriate for the hypothesis being tested?

-Is the sample size sufficient to ensure adequate power to address the hypothesis being tested?

-Were correct statistical analysis used to support conclusions?

-Are there concerns about ethical or regulatory requirements being met?

Reviewer #1: I am not a modeler but I assume methods are correct

Reviewer #2: It is necessary to provide a more detailed explanation of how the blocks, which served as the units of analysis are constructed. 

How big are they? Is there a lot of variability? Why were blocks chosen as the units of analysis?

To enhance the description of the study area, I recommend including a map of India highlighting the specific regions under investigation.

Furthermore, it is essential to provide a better explanation of how the priors were defined. Clarifying the process will help readers comprehend the underlying assumptions and establish a stronger foundation for the subsequent analyses.

Authors mention that restricted models or orthogonal analysis were not conducted due to computational constraints. It would be beneficial to explicitly outline these constraints, enabling readers to better evaluate the feasibility of such analyses. If conducting these analyses is genuinely impossible, please provide a thorough justification for this. On the contrary, such analyzes must be carried out.

Reviewer #3: The objectives are consistent with the stated hypotheses. 

The study design is appropriate and well detailed. It is a complementary study, adds temporal data and evaluation of variables that improve the results and applications on a larger scale.

The ethical standards are described and complied with. 

Add a general map where Bihar and Jharkhand are geographically located.

**Results**

-Does the analysis presented match the analysis plan?

-Are the results clearly and completely presented?

-Are the figures (Tables, Images) of sufficient quality for clarity?

Reviewer #1: Yes

Reviewer #2: In Table 1, it is observed that in 2020, some variables exhibit correlation coefficients that significantly deviate from those of previous years. It would be valuable to explain and discuss the reasons behind this discrepancy in detail.

-In Figure 1, please include a legend for the gray color used in the visual representation to ensure readers can accurately interpret the information presented.

Reviewer #3: The results are presented in a clear and orderly manner; figures, tables and supplementary material are sufficient.

**Conclusions**

-Are the conclusions supported by the data presented?

-Are the limitations of analysis clearly described?

-Do the authors discuss how these data can be helpful to advance our understanding of the topic under study?

-Is public health relevance addressed?

Reviewer #1: See summary and general comments

Reviewer #2: -The results regarding population density require a more in-depth discussion. Why is the incidence higher in less dense areas? Why is this the only significant variable in the models? Why did the other variables lose their significance? Although you briefly mention these points in the limitations section, I recommend further elaboration.

-It is crucial to include a discussion on data quality and whether potential biases could have influenced the study's findings

-Conclusions should offer more objective insights regarding the implications of the results for health interventions. Consider providing more explicit recommendations or directions for future actions based on the findings of your study.

Reviewer #3: There are no limitations beyond finding a single endemic factor, probably further investigation can be done with variables related to "urbanisation" or the possibility of including vector abundance from available dynamics studies. In relation to the likely biases in VL case reports, the COVID pandemic is cited as a drawback.

The authors emphasise that incorporating spatial, temporal and spatio-temporal interactions into the analysis led to an increase in the predictive level of the training and validation models, making it possible to adjust interventions in those blocks with greater spatial heterogeneity.

**Editorial and Data Presentation Modifications?**

Reviewer #1: (No Response)

Reviewer #2: (No Response)

Reviewer #3: (No Response)

**Summary and General Comments**

Reviewer #1: Modelling spatiotemporal patterns of visceral leishmaniasis incidence in India using

environment, bioclimatic and demographic data, 2013-2021

The authors have developed a model to predict incidence of VL at block level in Bihar and Jharkhand. They fitted data over the period 2013-2018 in a model incorporating environment, climatic and demographic factors. The model was validated with the monthly cases for 2019-2020, assessing the proportion of observations that fall inside the 95% credible interval for the predicted number of VL cases per month. During the training period model predictions agreed for > 93% with observed numbers of cases, over the testing period agreement went up to 99%. The authors conclude that the forecast can be used to monitor progress of VL elimination at the block level, to select blocks with incidence above the elimination threshold for targeted interventions and to prevent or reduce risk of resurgence in post-elimination settings.

Major comments:

This is a well-written manuscript but I would like to have some more clarity on the practical implications of using this model in the context of the VL elimination program in India. Outcomes are expressed in rates per 10,000 but what does that mean at block level? As numbers go further down, stochasticity will become more of an issue. Having 20 instead of 10 cases in a year at block level means a doubling of the rates but for program management purposes it may not be that much of a difference. 

When it comes to monitoring elimination and preventing resurgence, what does this model add to epidemiological surveillance? Either case based surveillance or serological surveillance would provide more reliable information for monitoring transmission. 

The authors selected 458 endemic blocks from Bihar as well as 11 non-endemic blocks and 33 endemic blocks from Jharkhand. How comparable were the non-endemic blocks from Bihar to endemic blocks? Were they rural blocks and where were they situated? And why were there no non-endemic blocks from Jharkhand included as in Jharkhand VL is much more limited in distribution. Would it not be interesting to see how the model performs when including non-endemic rural blocks of Jharkhand? This might be even more relevant in view of observations from Nepal where VL now pops up in areas where it never was before. 

Minor comments:

In line 374-375 it is stated that: ‘Consequently, the blocks with low population density tend to have high incidence of VL and vice versa’. This seems intuitive to me because VL is a rural disease. But on page 26 (no more line numbers there) the authors state that: ‘Our findings on the positive association of population density with VL incidence supports the earlier work in Teresina, Piauí State, Brazil that areas with high population density and abundant vegetation are associated with high VL incidence [64]’. Is this not contradictory? 

Data presented are until May, 2021. But we are two years later now and the KAMIS database is kept up to date. Would it not be possible to assess how well the predictions worked until the end of 2022 to get a stronger indication of how useful the model could be for forecasting VL incidence at block level?

Reviewer #2: The study provides valuable insights into the incidence of visceral leishmaniasis in specific regions of India. The topic is highly relevant and contributes to our understanding of the factors influencing the spatial and temporal evolution of the disease. 

However, I would like to offer some suggestions to improve the clarity and accessibility of your article. Other points that I consider relevant and essential for improving the study were described in the previous answers.

-Firstly, the author summary could benefit from a clearer and more accessible language to ensure readers less familiar with the subject matter can easily comprehend the content. Additionally, several sections of the text require additional explanatory information to enhance understanding for readers who may be less familiar with the specific analyses performed.

-Regarding the title, it would be beneficial to specify that the study focuses solely on two states of India, providing a more concise and accurate representation of the research scope.

Reviewer #3: The problem of spatial and temporal scale is a well-known and much-discussed issue when investigating an early warning and response system in the generation of leishmaniasis outbreaks. In general, phlebotomine sandfly species tend to be microfocal and it is imperative to have possible interventions at different spatial-temporal scales, which makes this type of approximation studies extremely essential to apply in public health in order to know the system of the area studied and to be able to go deeper into the different aspects where there is a lack of information.

PLOS authors have the option to publish the peer review history of their article (what does this mean?). If published, this will include your full peer review and any attached files.

Reviewer #1: No

Reviewer #2: No

Reviewer #3: No
---

## [Decision Letter · Decision Letter 1]

26 Jan 2024

Dear Dr. Subramanian,

We are pleased to inform you that your manuscript 'Modelling spatiotemporal patterns of visceral leishmaniasis incidence in two endemic states in India using environment, bioclimatic and demographic data, 2013-2022' has been provisionally accepted for publication in PLOS Neglected Tropical Diseases.

Best regards,

Alberto Novaes Ramos Jr

Academic Editor

Charles Jaffe

Section Editor

Reviewer's Responses to Questions

**Key Review Criteria Required for Acceptance?**

**Methods**

-Are the objectives of the study clearly articulated with a clear testable hypothesis stated?

-Is the study design appropriate to address the stated objectives?

-Is the population clearly described and appropriate for the hypothesis being tested?

-Is the sample size sufficient to ensure adequate power to address the hypothesis being tested?

-Were correct statistical analysis used to support conclusions?

-Are there concerns about ethical or regulatory requirements being met?

Reviewer #1: (No Response)

Reviewer #2: Yes

**Results**

-Does the analysis presented match the analysis plan?

-Are the results clearly and completely presented?

-Are the figures (Tables, Images) of sufficient quality for clarity?

Reviewer #1: (No Response)

Reviewer #2: Yes

**Conclusions**

-Are the conclusions supported by the data presented?

-Are the limitations of analysis clearly described?

-Do the authors discuss how these data can be helpful to advance our understanding of the topic under study?

-Is public health relevance addressed?

Reviewer #1: (No Response)

Reviewer #2: Yes

**Editorial and Data Presentation Modifications?**

Reviewer #1: (No Response)

Reviewer #2: (No Response)

**Summary and General Comments**

Reviewer #1: (No Response)

Reviewer #2: All comments have been addressed!

PLOS authors have the option to publish the peer review history of their article (what does this mean?). If published, this will include your full peer review and any attached files.

Reviewer #1: No

Reviewer #2: **Yes: **Vinícius Silva Belo

---

## [Editor Report · Acceptance letter]

30 Jan 2024

Dear Dr. Subramanian,

We are delighted to inform you that your manuscript, "Modelling spatiotemporal patterns of visceral leishmaniasis incidence in two endemic states in India using environment, bioclimatic and demographic data, 2013-2022 ," has been formally accepted for publication in PLOS Neglected Tropical Diseases.

Best regards,

Shaden Kamhawi

co-Editor-in-Chief

Paul Brindley

co-Editor-in-Chief
